



# Observed Process-level Constraints of Cloud and Precipitation Properties over the Southern Ocean for Earth System Model Evaluation

McKenna W. Stanford[1,2], Ann M. Fridlind[2], Israel Silber[3], Andrew S. Ackerman[2], Greg Cesana[1,2], Johannes Mülmenstädt[4], Alain Protat[5,7], Simon Alexander[6,7], and Adrian McDonald[8]

[1]Center for Climate Systems Research, Columbia University, New York, NY, USA

[2]NASA Goddard Institute for Space Studies, New York, NY, USA

[3]Department of Meteorology and Atmospheric Science, Pennsylvania State University, University Park, PA, USA

[4]Atmospheric Sciences & Global Change Division, Pacific Northwest National Laboratory, Richland, WA, USA

[5]Australian Bureau of Meteorology, Melbourne, VIC, Australia

[6]Australian Antarctic Division, Kingston, TAS, Australia

[7]Australian Antarctic Partnership Program, Institute for Marine and Antarctic Studies, University of Tasmania, Hobart, TAS, Australia

[8]School of Physical and Chemical Sciences, University of Canterbury, Christchurch, New Zealand

**Correspondence:** McKenna W. Stanford (mws2175@columbia.edu)

**Abstract.**

Over the remote Southern Ocean, cloud feedbacks contribute substantially to Earth system model (ESM) radiative biases. The evolution of low Southern Ocean clouds (cloud top heights $< \sim 3$ km) is strongly modulated by precipitation and/or evaporation, which act as the primary sink of cloud condensate. Constraining precipitation processes in ESMs requires robust observations suitable for process-level evaluations. A year-long subset (April 2016 – March 2017) of ground-based profiling instrumentation deployed during the Macquarie Island Cloud and Radiation Experiment (MICRE) field campaign (54.5 °S, 158.9 °E) combines a 95 GHz (W-band) Doppler cloud radar, two lidar ceilometers, and balloon-borne soundings to quantify the occurrence frequency of precipitation from liquid-phase cloud base. Liquid-based clouds at Macquarie Island precipitate $\sim 70\%$ of the time, with deeper and colder clouds precipitating more frequently and at a higher intensity compared to thinner and warmer clouds. Supercooled cloud layers precipitate more readily than layers with cloud top temperatures $> 0\,°C$, regardless of the geometric thickness of the layer, and also evaporate more frequently. We further demonstrate an approach to employ these observational constraints for evaluation of a 9-year GISS-ModelE3 ESM simulation. Model output is processed through the Earth Model Column Collaboratory (EMC²) radar and lidar instrument simulator with the same instrument specifications as those deployed during MICRE, therefore accounting for instrument sensitivities and ensuring a coherent comparison. Relative to MICRE observations, the ESM produces a smaller cloud occurrence frequency, smaller precipitation occurrence frequency, and greater sub-cloud evaporation. The lower precipitation occurrence frequency by the ESM relative to MICRE contrasts with numerous studies that suggest a ubiquitous bias by ESMs to precipitate too frequently over the SO when compared with satellite-based observations, likely owing to sensitivity limitations of space-borne instrumentation and different sampling methodologies for ground- versus space-based observations. Despite these deficiencies, the ESM reproduces the observed





tendency for deeper and colder clouds to precipitate more frequently and at a higher intensity. The ESM also reproduces specific cloud regimes, including near-surface clouds that account for $\sim 25\%$ of liquid-based clouds during MICRE and optically thin, non-precipitating clouds that account for $\sim 27\%$ of clouds with bases higher than 250 m. We suggest that the demonstrated framework, which merges observations with appropriately constrained model output, is a valuable approach to evaluate processes responsible for cloud radiative feedbacks in ESMs.

## 1  Introduction

Extratropical shortwave (SW) radiation cloud feedbacks are a significant source of uncertainty in Earth system model (ESM) projections of a perturbed climate (e.g., Caldwell et al., 2016; McCoy et al., 2020). In particular, ESMs in phase 5 of the Coupled Model Intercomparison Project (CMIP5; Taylor et al., 2012) exhibit high-biased SW absorption due to a deficit in low- and mid-level cloudiness over the Southern Ocean (SO) (Bodas-Salcedo et al., 2014, 2016; Naud et al., 2014). Low-
level cloud ($< \sim 3$ km) that forms in the warm and cold sectors of extratropical cyclones accounts for up to 80% of annual fractional cloud cover in observations (Mace et al., 2009). Cloud condensate amount and sustenance are heavily modulated by precipitation (Kay et al., 2016b; Tan et al., 2016), which is the dominant factor of moisture depletion (McCoy et al., 2020). In a warming climate, an expected shift to more liquid-bearing ("warm") clouds has been shown to increase liquid-phase cloud amount, increase optical depth, and contribute to a larger negative cloud feedback (Mitchell et al., 1989; Tsushima et al., 2006;
Mülmenstädt et al., 2021), following from findings that precipitation efficiency is generally weaker in warm clouds compared to supercooled clouds (Mitchell et al., 1989; Senior and Mitchell, 1993; Tsushima et al., 2006; Hoose et al., 2008). Properly predicting extratropical SW cloud feedbacks is thus dependent on an ESM's ability to faithfully represent both observed precipitation occurrence frequency and cloud phase, but these are common shortcomings of ESMs, especially over the SO (Kay et al., 2016b, 2018; Naud et al., 2020; Gettelman et al., 2020; Cesana et al., 2022).

Robust observational constraints are needed in order to understand precipitation occurrence frequency in ESMs. Space-borne platforms offer the longest and most spatially expansive constraints but have some limitations. For example, the CloudSat Cloud Profiling Radar (CPR; Stephens et al., 2002) experiences contamination in the lowest 1 km due to ground clutter that hinders detection of low marine clouds, inducing a miss rate of up to 39% over the global oceans (Liu et al., 2016; McErlich et al., 2021). Low CPR sensitivity also limits detection of optically thin clouds and its relatively coarse horizontal resolution misses
shallow cumulus clouds (Rodts et al., 2003; Zhang and Klein, 2013; Cesana et al., 2019a). Lamer et al. (2020a) found that CPR limitations impeded detection of warm marine boundary layer clouds over the eastern North Atlantic by 29%-43% and distorted cloud macroscopic properties compared to ground-based instrumentation. Over the Arctic and Antarctic, Silber et al. (2021) found that differences in sensitivity and precipitation detection algorithms can reduce space-borne estimates of cloud-base and surface precipitation occurrence frequency by more than 50%. For the purpose of cloud base precipitation evaluation,
space-based lidars furthermore become attenuated in visibly opaque layers with optical depths $> \sim 3$, preventing identification of a cloud layer throughout the entire column and thus leaving cloud base height poorly defined (Vaughan et al., 2009).



Another approach used for characterizing precipitation frequency and intensity is the use of ground-based remote sensing deployments that allow for long-term (order of months to years) statistics to be compiled at high temporal and vertical spatial resolution (Illingworth et al., 2007; Bühl et al., 2016; Ansmann et al., 2019; Bühl et al., 2019; Lamer et al., 2020b; Griesche et al., 2021; Ramelli et al., 2021; Silber et al., 2021). Such ground-based datasets usually include periodic balloon soundings that provide direct colocated measurements of atmospheric thermodynamic state, which are generally missing from satellite remote sensing. Although often horizontally limited (employing only zenith-viewing instruments), such methods provide a means to obtain characteristics of shallow, boundary-layer limited clouds that are regionally ubiquitous that is complementary to satellite remote sensing. For instance, Silber et al. (2021) used measurements from Utqiaġvik (formerly Barrow), North Slope of Alaska (NSA;  Verlinde et al., 2016) and McMurdo Station, Antarctica (Lubin et al., 2020b) to establish the precipitation occurrence frequency in polar supercooled clouds. Using a combined sounding-radar approach, they found that 85% (75%) of supercooled cloud layers are precipitating from liquid cloud base at the NSA (McMurdo Station). Lamer et al. (2020b) similarly used a combined radar-lidar approach at the U.S. Department of Energy (DOE) Atmospheric Radiation Measurement (ARM) program's Eastern North Atlantic (ENA) site to determine that 80% of warm clouds in subsidence regimes are precipitating from cloud base. Ship-based deployments have also been extensively evaluated using these profiling measurement techniques. For example, Griesche et al. (2021) combined ship-based lidar, radar, and radiosondes during an Arctic summer voyage and found that for cloud top temperatures > -15 °C, surface-coupled clouds were more likely to contain ice than were surface-decoupled clouds. These techniques have also been used to perform mixed-phase microphysical retrievals, such as ice- and liquid-mass flux (Bühl et al., 2016) and ice crystal number concentrations (Bühl et al., 2019).

Addressing ESM biases over the SO has recently motivated numerous airborne and ship-based field campaigns to characterize cloud, aerosol, and radiation properties across a latitudinal band from $\sim 45 - 75$ °S (Mace and Protat, 2018a, b; Kremser et al., 2021; McFarquhar et al., 2021). Ship-based campaigns equipped with lidar, radar, and radiosondes have yielded results on cloud processes and microphysics (Mace and Protat, 2018a, b; McFarquhar et al., 2021). For example, clouds near the Antarctic coast were found to have higher droplet number concentrations than those further north due to continental air masses with large cloud condensation nuclei concentrations and increased sulfate aerosol (Mace et al., 2021), and supercooled liquid drizzle is often observed beneath clouds in the same coastal Antarctic region (Alexander et al., 2021).

Complementary to these ship-based campaigns, the Macquarie Island Cloud and Radiation Experiment (MICRE) was organized by the DOE ARM program, the Australian Bureau of Meteorology (BoM), and the Australian Antarctic Division (AAD) from March 2016 to March 2018. MICRE is thus far the only stationary, ground-based campaign to provide an annual cycle of SO cloud measurements at a fixed site (where the SO is defined broadly as 45 to 75 ° S). Situated at 54.5 °S and 158.9 °E, Macquarie Island is well located in the middle of the SO midlatitude storm track, making it a valuable location to observe cloud regimes responsible for ESM biases and has been subject to detailed study (e.g., Adams, 2009; Wang et al., 2015; Lang et al., 2018, 2020; Tansey et al., 2022). Tansey et al. (2022) combined data streams from a surface disdrometer, cloud radar, and tipping bucket rain gauge during MICRE and found that surface precipitation occurs $44 \pm 4\%$ of the time and is dominated by relatively small particles (< 1 mm in diameter). Wang et al. (2015) evaluated an 8-year record (2003-2011) of 3-hourly tipping bucket rain gauge observations at Macquarie Island with a lower measurement limit of 0.2 mm hr$^{-1}$ and found that surface




precipitation occurred 36% of the time with a large contribution from light precipitation rates. Lang et al. (2020) used 18 years of hourly surface precipitation measurements to reveal a diurnal cycle in precipitation that peaks during night/early morning and is strongest during Austral summer.

In this work we report a combined analysis of measurements from a 95 GHz (W-band) zenith-pointing Doppler cloud radar, two lidar ceilometers, and atmospheric soundings deployed at Macquarie Island that were coincident during a year of the MICRE campaign (April 2016 to May 2017; McFarquhar et al., 2021; Tansey et al., 2022). A leading objective is to merge instrument data streams to compute the precipitation occurrence frequency from liquid cloud base (LCB). A focus on LCB precipitation, whether or not the precipitation reaches the surface, provides an important constraint for ESMs because it means

that an active precipitation process is occurring that should be represented by a given model's physics parameterizations. In an observational analysis of coalescence scavenging over the SO, Kang et al. (2022) found that light precipitation rates ($<$ 0.1 mm hr$^{-1}$) have a significant impact on scavenging of cloud condensation nuclei and the resulting cloud droplet number concentration, demonstrating the relevance of precipitation rates at the low-intensity limit. Moreover, understanding the degree to which evaporation or sublimation is prevalent below cloud base is important as it impacts sub-cloud precipitation accumu-

lation, boundary layer structure, and cloud mesoscale organization. For example, Heymsfield et al. (2020) used satellite-based radar measurements to evaluate hydrometeor phase contributions to the global precipitation budget and found a significant contribution from evaporation of melted, frozen precipitation in an ESM. Retrievals of LCB precipitation rates, cloud top and base temperatures, and cloud geometric thickness are used here to investigate the degree to which LCB precipitation properties are sensitive to the cloud top supercooling and the cloud geometric thickness. Retrievals of precipitation occurrence frequency are

then projected onto sensitivities that emulate instrument and algorithm sensitivity, providing comparative uncertainties associated with space-based retrievals that can be used going forward to inform strategies for fusion of ground- and satellite-based data sources for model evaluation.

     The merged MICRE dataset is finally used to evaluate a 9-year ESM simulation by means of the Earth Model Column Collaboratory (EMC$^2$; Silber et al., 2022) radar and lidar instrument simulator and subcolumn generator. EMC$^2$ was designed to

enable robust comparisons between ground-based observations and ESM column physics in a manner that remains faithful to the model's physics assumptions. Using EMC$^2$, forward simulations are performed on ESM output from the National Aeronautics and Space Administration (NASA) Goddard Institute for Space Studies (GISS) ModelE3 (GISS-ModelE3; Cesana et al., 2019b, 2021) ESM at 2.0 × 2.5 ° resolution as a free-running global simulation with prescribed sea surface temperatures and sea ice distributions. Vertical profiles of microphysical quantities required for forward simulation of remote-sensing observ-

ables are output at time-step frequency at Macquarie Island's geographic location and processed through EMC$^2$ to produce radar and lidar calculations consistent with the specifications of instrumentation deployed during MICRE. In this manner, we demonstrate a framework for process-level evaluation of ESM column physics against long-term, ground-based observations over the SO using the MICRE measurements.

     The remainder of the article is structured as follows: data and methods, including observational datasets and precipitation

detection algorithm development, are described in Sect. 2. Observational results are presented in Sect. 3, and a demonstration



of GISS-ModelE3 evaluation against those results is provided in Sect. 4. Implications of findings for ESMs, satellite retrievals, and designing future SO missions are presented in Sect. 5, and conclusions are summarized in Sect. 6.

## 2 Data and Methods

### 2.1 Data

Instruments used in this study include the BoM's Bistatic Radar System for Atmospheric Studies (BASTA; Delanoë et al., 2016) 95 GHz (W-band) zenith-pointing Doppler cloud radar, ARM's Vaisala CT25K 910 nm ceilometer (Morris et al., 2016; Morris, 2016), the University of Canterbury's Vaisala CL51 910 nm ceilometer (Alexander and McDonald, 2019), and 12-hourly atmospheric balloon soundings conducted by the Australian Bureau of Meteorology (Barnes-Keoghan, 2000). A 2-hour example of data from this instrumentation is shown in Fig. 1.

The BASTA radar operates in four three-second modes with varying sensitivity and vertical resolution. Here, we use the 25-m mode most suitable for detecting low-level liquid cloud layers (Delanoë et al., 2016), for which the effective temporal resolution is 12 seconds with a vertical range from 125 m to 12 km above ground level (AGL). Although MICRE extended over 2 years (2016 to 2018), the BASTA radar's residence was limited to only approximately the first year of the campaign (April 2016 to March 2017). Calibration of BASTA is achieved using recent ship-based campaign data from BASTA, a 24 GHz

Micro-Rain Radar PRO, an optical disdrometer, and T-matrix calculations (Protat et al., 2019). BASTA has a sensitivity of -36 dBZ at 1 km AGL and any bins with values below the theoretical minimum reflectivity ($Z_{e,\min}$, see Appendix B) are treated as free of hydrometeors.

    The ARM ceilometer has native 16-second temporal resolution and 10-m vertical resolution extending from the surface to 7.7 km above ground level (AGL). The cloud base height (CBH) product (Morris, 2016) allows the detection of up to three

CBHs, but only the lowest identified CBH is used here. Uncertainty in CBH by the ARM ceilometer is $\pm 5$ m (half of the native vertical resolution; Morris, 2016). The University of Canterbury ceilometer has native 6-second temporal resolution and 10-m vertical resolution with three CBHs retrieved up to 15.4 km AGL at 10-m resolution. The ARM ceilometer is primarily used for CBH detection, though due to prolonged blackout periods, the University of Canterbury ceilometer is used to fill in gaps when the ARM ceilometer was not operational. Because the highest identifiable CBH by the ARM ceilometer is 7.7 km AGL,

all CBHs higher than 7.7 km identified by the University of Canterbury ceilometer are discarded, though this limit is high enough to encapsulate the overwhelming majority of liquid layers. CBH detections used from both ceilometers come from the vendor's proprietary software, which is generally associated with a peak signal in attenuated backscatter. Collectively, the merged ceilometer dataset is referred to as CEIL.

    Soundings were released nominally every 12 hours and measured atmospheric pressure, temperature, and relative humidity with respect to liquid water ($RH_{liq}$). Uncertainties in $RH_{liq}$, temperature, and pressure are assumed to be 5%, 0.5 °C, and 1

hPa, respectively (Holdridge, 2020). A surface meteorology station is also used contextually in our analysis (Howie and Protat, 2016).

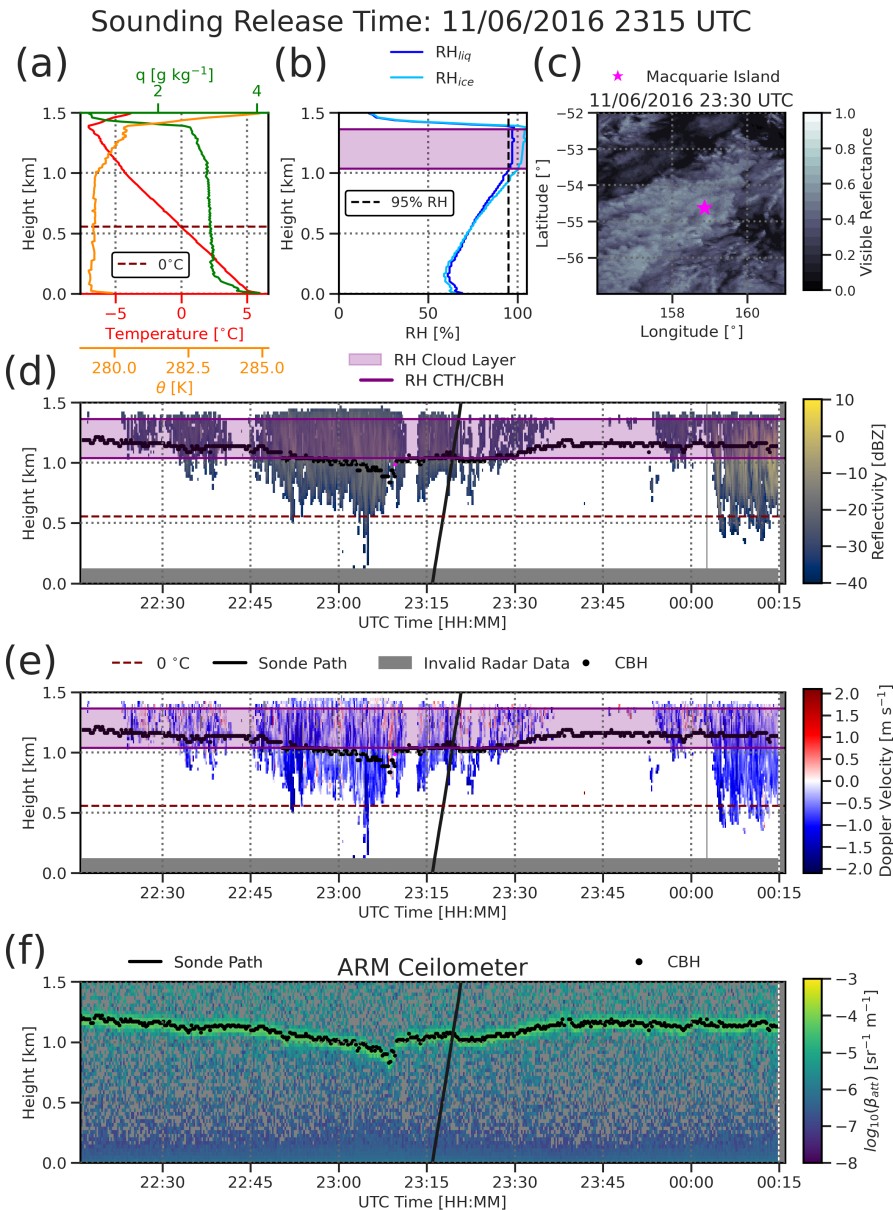

**Figure 1.** Two-hour example of measurements at Macquarie Island: (a) sounding temperature, water vapor mixing ratio ($q$), and potential temperature ($\theta$) with melting level indicated (dashed line), (b) relative humidity with respect to liquid water (RH$_{liq}$) and ice (RH$_{ice}$) with 95% RH$_{liq}$ indicated (dashed line), (c) satellite visible reflectance from the Himawari-8 satellite (ARM User Facility, 2016) and the location of Macquarie Island, (d) BASTA radar reflectivity, (e) BASTA mean Doppler velocity, and (f) ARM ceilometer attenuated backscatter ($\beta_{att}$). In panels (d)-(f), the sounding path is shown as a black line from 2315 UTC and the cloud base heights (CBHs) are shown as black dots. Purple shading in panels (b), (d), and (e) indicates the vertical extent where sounding RH$_{liq}$ > 95%.



## 2.2 Methods

All instruments are merged and gridded onto the BASTA time-height grid of 12 seconds and 25 m, and time periods with invalid
radar and/or ceilometer data are discarded. Cloud-base precipitation occurrence frequency depends on the CEIL-identified
CBH, the uncertainties for which are discussed next along with calculations of cloud macrophysical and thermodynamic
properties. Derivations of cloud-base and surface precipitation occurrence frequency ($P_\mathrm{cb}$ and $P_\mathrm{sfc}$, respectively) are then
described, followed by retrievals of cloud-base precipitation rates ($R_\mathrm{cb}$). Appendix A provides a list of abbreviations and
notation used throughout the manuscript.

### 2.2.1   Cloud Macrophysics and Thermodynamics

All CBH detections by CEIL are assumed to be liquid cloud base (LCB) heights. Silber et al. (2018) compared various LCB
height products for polar supercooled liquid cloud cases and found that the ARM ceilometer occasionally detects liquid cloud
bases that are actually ice as identified by polarization lidar data, but these false detections remain below 2% of the distribution
for any given altitude, though we note the vastly different environments sampled between Macquarie Island and the polar
sites they evaluated. We also note that although a polarization lidar was present during the MICRE campaign, post-processing
limitations prevent its use in a statistical manner.

Additionally, Silber et al. (2018) found based on a comparison with high-spectral resolution lidar (HSRL) measurements
that, on average, the ARM ceilometer detects LCB 36-50 m in-cloud (site-dependent), but that it performs well in regions of
heavy precipitation and exhibits low variability compared to other CBH detection algorithms. Sensitivity to biases in CBH are
evaluated in Appendix C by decreasing the CBH by 25 to 50 m (i.e., one to two BASTA bins) for all retrievals. Herein, we also
discard any CBH detections that have a cloud base temperature (CBT) colder than the homogeneous freezing level (taken to
be -38 °C).

In fog, CEIL signals attenuate completely near the surface, such that a CBH is identified near the surface and most often at
altitudes below 250 m. Since $P_\mathrm{cb}$ is evaluated at a minimum height that is at least 200 m AGL based on radar ground clutter
blocking, these fog-influenced backscatter profiles contribute minimally ($< 3$%) to profiles used for precipitation detection. For
CBHs $< 250$ m, where they are relatively common, these CEIL backscatter profiles indicative of fog are flagged and discussed
separately in Sect. 3.4.2 and Appendix F.

Independent evaluation of CEIL LCB was made by using in-situ $RH_\mathrm{liq}$ thresholds from soundings. CEIL-recognized LCBs
at sounding release times were colocated and are shown as a function of $RH_\mathrm{liq}$ and temperature in Fig. D1, indicating that more
than 66 (80)% of CEIL-recognized LCBs exhibit $RH_\mathrm{liq} > 95$ (90)%. Silber et al. (2020a) found that $> 90$% of polar supercooled
cloud bases identified by an HSRL had concurrent sounding $RH_\mathrm{liq} > 95$%. The reduced percentage of CEIL-recognized LCBs
with $RH_\mathrm{liq} > 95$% in the MICRE dataset compared to polar supercooled cloud layers in Silber et al. (2020a) can be attributed
at least in part to spatiotemporal discrepancies between the cloud environments sampled by the soundings and by CEIL. For
example, Fig. 1b shows that $RH_\mathrm{liq}$ drops quickly below the sounding-recognized LCB (i.e., where $RH_\mathrm{liq}$ first exceeds 95% in
purple shading). Therefore, variability in CBH by even 100 m (which is within the range of variability of CEIL CBHs for the



2-hour time period in Fig. 1f) can lead to RH$_\text{liq}$ < 95% at the CEIL-recognized LCB. In addition, there are frequently scenarios in which the sounding balloon passes in between horizontally inhomogeneous cloud layers, such that the sounding RH$_\text{liq}$ never reaches 95% despite the identification of nearby cloud via ceilometer. The approach taken by Silber et al. (2021) in which cloud boundaries were identified by sounding RH$_\text{liq}$ thresholds rather than lidar and radar was motivated by the prevalence of overcast multi-layer supercooled clouds in their polar cloud regimes and also enabled a sufficiently long sounding dataset over $\sim$ 7 years in the Arctic. By contrast, the relatively short duration of MICRE and the greater heterogeneity of cloud boundaries over the SO relative to polar clouds in our case motivates LCB identification via remote sensing instrumentation with higher temporal resolution (i.e., CEIL and BASTA). Although there remains uncertainty in LCB height identification, particularly due to unknowns regarding CBH algorithms, we have attempted to mitigate these uncertainties by evaluating CEIL LCBs against sounding RH$_\text{liq}$ measurements (Appendix D), accounting for fog-influenced CEIL profiles (Appendix F), and accounting for uncertainty in $P_\text{cb}$ due to errors in the height of LCB identified by CEIL (Appendix C). Potential improvements to instrument strategies for LCB height determination in future campaigns are also discussed below.

Cloud top height (CTH) is determined as the height at which a contiguous layer of reflectivity ($Z_e$) above the CEIL-identified cloud base drops below $Z_{e,\text{min}}$ (i.e., becomes free of hydrometeors). The difference between CTH and CBH defines the cloud geometric thickness. Cloud top temperature (CTT) and cloud base temperature (CBT) are determined by near-in-time atmospheric soundings. Soundings released at nominally 12-hr intervals are linearly interpolated onto constant altitude levels in order to form a continuous curtain plausibly consistent with the radar and CEIL measurements. During periods when soundings were released more than 12 hours apart, temperature is taken to be constant for 6 hours on either side of the sounding release time and time periods greater than 6 hours from the sounding release time are discarded, though we note that the results here are not sensitive to the time period surrounding a given sounding (not shown). During periods of robust stratiform precipitation, the interpolated 0 °C isotherm is found to be consistent with a melting layer or "bright band" (i.e., a steep increase in Doppler velocity and an apparent jump in radar reflectivity, see Austin and Bemis, 1950), further indicating relatively robust measurements of tropospheric temperature despite the coarse time frequency of measurements. Using CBT and CTT, cloud layers are subdivided into supercooled layers ( CBT and CTT < 0 °C), partially supercooled layers (CBT $\geq$ 0°C and CTT < 0 °C), and warm layers (CBT and CTT $\geq$ 0°C).

### 2.2.2 Precipitation Occurrence Frequency

Precipitation identification is determined by linearly averaging the reflectivity factor within a prescribed number of bins below the ceilometer-identified LCB height ($D_\text{min}$). Precipitation occurrence requires that the linearly averaged reflectivity exceeds the theoretical reflectivity minimum as a function of height ($Z_{e,\text{min}}$; Fig. B1) and that the minimum mean Doppler velocity within the range of bins is negative (downward, thus excluding updrafts). We note instances in which there exists a CEIL-identified CBH without coincident reflectivity, where the higher sensitivity of the ceilometer to smaller hydrometeors produces detectable backscatter returns from small droplets unregistered by the radar. We consider these instances to be non-precipitating clouds, which are discussed in detail in Sect. 3.4.1.





Cloud-base precipitation occurrence frequency ($P_{\mathrm{cb}}$) is calculated for varying minimum $Z_{e,\mathrm{min}}$ that ranges from -55 to 15
220    dBZ and for varying depths below cloud base ($D_{\mathrm{min}}$) used for reflectivity averaging, ranging from 50 m to 600 m. The minimum
detectable height of the radar ($h_{\mathrm{min}}$) is set to 150 m based on careful analysis of ground clutter contamination. The minimum
allowable CBH is thus $h_{\mathrm{min}} + D_{\mathrm{min}}$, ranging from 200 m to 750 m AGL depending on $D_{\mathrm{min}}$ (see Appendix E). Precipitation
occurrence frequency at the surface ($P_{\mathrm{sfc}}$) is also derived by linearly averaging reflectivity within a prescribed number of bins
above $h_{\mathrm{min}}$.

### 2.2.3    Precipitation Rates

Calculations of cloud-base precipitation rates ($R_{\mathrm{cb}}$) are determined by first identifying the temperature of LCB . For CBTs $\geq$
0 °C, the drizzle reflectivity-rain rate relationship ($Z$-$R$) from Comstock et al. (2004) is used ($Z = aR^b$, where $a = 25$ and $b =$
1.3). For CBTs $< 0$ °C, we follow the methodology of Silber et al. (2021) and Bühl et al. (2016), and use the Hogan et al. (2006)
parameterization for computing ice water content (IWC) via reflectivity and temperature and then compute ice water flux by
230    multiplying IWC by the minimum mean Doppler velocity within a prescribed depth below LCB ($D_{\mathrm{min}}$). This method assumes
the column beneath the LCB is subsaturated (supersaturated) with respect to liquid (ice). We note that there are significant
uncertainties related to these precipitation rate retrievals, especially considering a lack of robust $Z$-$R$ relationships derived for
SO clouds and the inability to robustly determine hydrometeor phase with the available instrumentation (e.g., Silber et al.,
2020b). Whereas Silber et al. (2021) found that $Z_e$ below LCB nearly universally increases downward in polar supercooled
235    cloud layers, indicative of ice-phase precipitation that grows by vapor diffusion below LCB (see their Appendix E), here we find
that only $\sim$ 45 to 60% of supercooled layers exhibit $Z_e$ increasing below LCB (not shown). This suggests that a relatively large
fraction of supercooled LCBs are precipitating primarily in the liquid phase, with warmer CTTs showing a greater likelihood for
decreasing $Z_e$ below LCB (indicative of evaporation). The presence of liquid-phase precipitation below a supercooled LCB is
consistent with Mace and Protat (2018a), who found that about half of supercooled liquid-based clouds contained liquid-phase
240    precipitation during the month-long ship-based Clouds, Aerosols, Precipitation, Radiation, and Atmospheric Composition over
the Southern Ocean (CAPRICORN I) campaign south of Tasmania (latitudinal range from $\sim$ 43 to 53 °S) from 13 March to
15 April 2016. Although there is uncertainty in the phase of precipitation and thus the retrieval used to derive $R_{\mathrm{cb}}$, we accept
these uncertainties as a starting point in this study and focus on quantifying trends as a function of cloud properties that are
expected to be important modulating factors.

### 245    3    Results

Liquid cloud bases are identified by CEIL in 76% of valid profiles in the merged MICRE dataset spanning nearly 1 year, with
month-to-month variability of $\sim$ 10% (not shown). Given this variability and only a single annual cycle, we do not evaluate
cloud and precipitation seasonal distributions but refer to Tansey et al. (2022) for a robust evaluation of MICRE's seasonal
cycle of surface precipitation. However, we note that this total cloud occurrence frequency matches that determined by Mace
250    and Protat (2018a) (76%) during the CAPRICORN I voyage.





CEIL is obscured 2.5% of the time, in which the ceilometer experienced attenuated backscatter but a cloud base could not be determined. These profiles are omitted from further analysis, though we note that obscuration commonly occurs during heavy precipitation or fog events, such that this 2.5% may be considered an uncertainty in total cloud occurrence frequency.

When an LCB was identified, 26% of identified LCBs are below 250 m AGL and are discussed in Sect. 3.4.2. The remaining 74% of LCBs are above 250 m AGL and are used for precipitation detection. Of these, 61% of layer LCBs are supercooled (i.e., CBT $< 0°$C). Precipitation occurrence frequencies are discussed next.

### 3.1 Cloud-base Precipitation Occurrence Frequency ($P_{cb}$)

Cloud-base precipitation occurrence frequency ($P_{cb}$) is first discussed in terms of the depth below cloud base used for precipitation detection ($D_{min}$, equivalent to the vertical resolution) and the minimum reflectivity threshold ($Z_{e,min}$; Fig. 2). As in Silber et al. (2021), this approach simultaneously illustrates both the MICRE dataset characteristics (in the lower left-hand corners in Fig. 2 panels) and quantities roughly comparable to a wide range of current and future satellite instrument characteristics. For example, the $Z_{e,min}$ and $D_{min}$ sensitivities of the CloudSat 2C-Precip-Column (2C-PC; Haynes et al., 2009) and 2C-Snow-Column (Wood et al., 2014) "possible" and the 2C-PC "certain" data products are shown as symbols in Fig. 2. At the BASTA $Z_{e,min}$ sensitivity and $D_{min} = 50$ m, 69% of clouds are precipitating from LCB (Fig. 2a) and decreases as both a function of $D_{min}$ and $Z_{e,min}$.

Supercooled layer $P_{cb}$ for BASTA is 61% (Fig. 2c) and warm layer $P_{cb}$ is 66% (Fig. 2g). While supercooled $P_{cb}$ is not a strong function of $D_{min}$, warm layer $P_{cb}$ decreases by a factor of 2 in the range of $D_{min}$ shown. In subsaturated air below LCB, liquid-phase cloud drops are expected to evaporate. As $D_{min}$ increases and $Z_e$ is averaged over a larger depth, evaporating drops become smaller such that the average $Z_e$ drops below the radar sensitivity. This is demonstrated at the surface (Fig. 2h), whereby the precipitation occurrence decreases by 12 percentage points relative to cloud base. Conversely, the sub-cloud environment for supercooled layers precipitating in the ice phase is expected to be supersaturated with respect to ice (though temperature-dependent), allowing for ice growth via vapor deposition and thus increasing $Z_e$ below LCB (Silber et al., 2021). The neutral slope of supercooled $P_{cb}$ as a function of $D_{min}$ indicates precipitation that is not strictly growing in the ice phase nor evaporating in the liquid phase. As described above, $Z_e$ below LCB was found to often *decrease* below LCB, indicating that a fraction of these supercooled cloud layers are precipitating primarily in the liquid phase, but the influence of precipitating ice is present. Indeed, near the surface, supercooled precipitation occurrence frequency ($P_{sfc}$) decreases by 19 percentage points (Fig. 2d), suggesting either evaporating liquid-phase precipitation from supercooled cloud layers, sublimation of ice, or evaporation of melted ice precipitation. Evaporation is discussed in more detail in Sect. 3.3.

Partially supercooled $P_{cb}$ is 97% for BASTA (Fig. 2e) and decreases by only 7 percentage points near the surface (Fig. 2f). These partially supercooled layers are shown below to generally be much thicker compared to purely supercooled or warm cloud layers and also to precipitate at a higher intensity, both of which are a likely reason for higher $P_{cb}$ and less evaporation. Finally, we note that sensitivities of $P_{cb}$ to potential biases in LCB height as discussed by Silber et al. (2018) are addressed in Appendix C and Fig. C1.



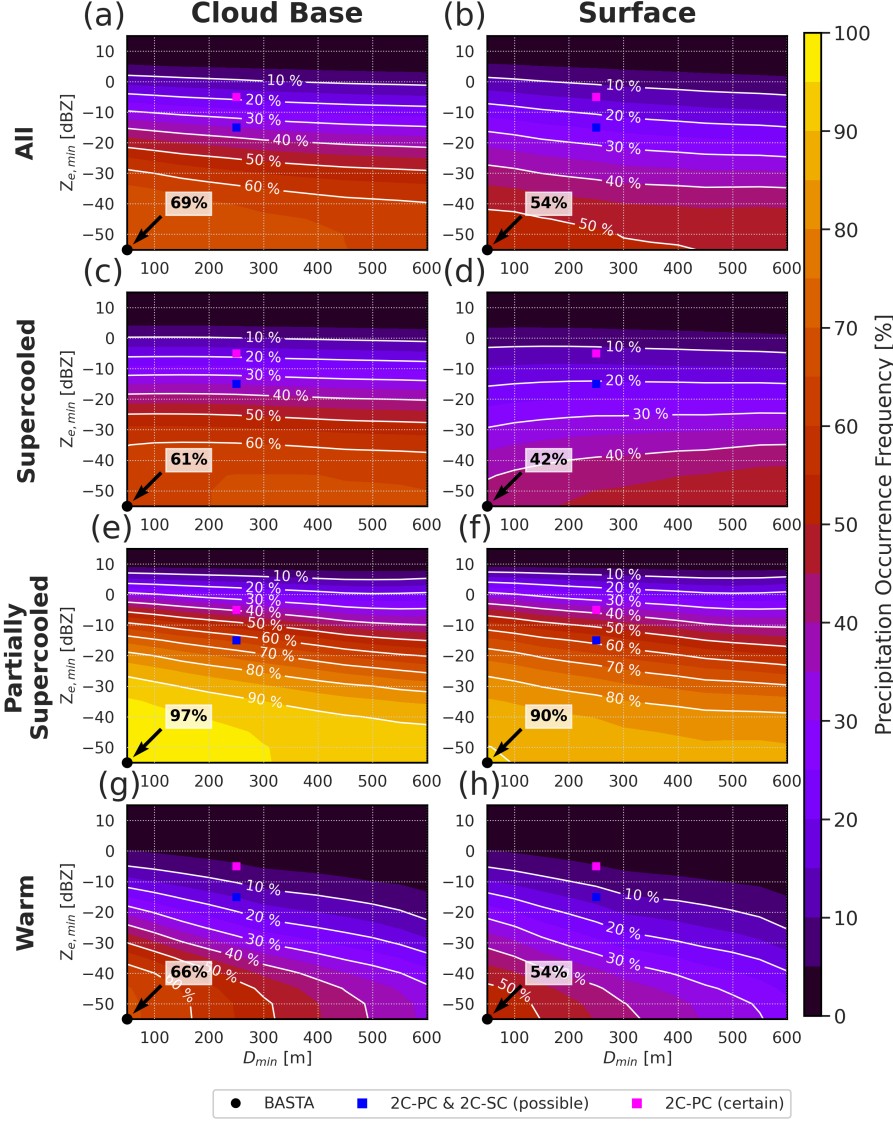

**Figure 2.** Precipitation occurrence frequency ($P_{cb}$, contours and color fill) as a function of the minimum reflectivity threshold ($Z_{e,min}$; ordinate) and the depth below cloud base used to detect precipitation ($D_{min}$; abscissa). All cloud layers are shown in the top row, supercooled layers in the second row, partially supercooled layers in the third row, and warm layers in the bottom row. The first column is for precipitation from cloud base ($P_{cb}$) and the second column is for precipitation at the surface ($P_{sfc}$). The black circles in the bottom left-hand corner of each panel represent the BASTA $Z_{e,min}$ and $D_{min}$ = 50 m (2 range gates). Blue and magenta symbols on all plots represent the $Z_{e,min}$ and $D_{min}$ (i.e., the vertical resolution) of the CloudSat 2C-PC/2C-SC "possible" and 2C-PC "certain" data products.

The projection of $P_{cb}$ is onto cloud thermodynamics and macrophysics is performed hereafter assuming a constant $D_{min}$

285 = 100 m (4 range gates) to limit artifacts from false detections, and the native BASTA $Z_{e,min}$ profile is retained. Occurrence



frequencies and the precipitating fraction of cloud layers are shown as a function of cloud thickness, CBH, and CTT in Fig. 3, where occurrence frequencies are normalized by all cloud layers (pink) and by non-precipitating cloud layers (green) and the precipitating fraction is calculated for all samples in a given cloud property bin. Non-precipitating cloud layers are thinner (Fig. 3a-d) and CBHs are higher (Fig. 3e-h) relative to all cloud layers, and the precipitating fraction increases with increasing

290 cloud thickness and decreases with increasing CBH. Partially supercooled cloud layers are generally thicker and CBHs are lower relative to purely supercooled layers. Cloud thickness and CBH distributions for all layers follow closely the supercooled layer distributions, consistent with Fig. E1 showing that the majority of cloud layers are supercooled.

Cloud layers with CTTs < -20 °C (Fig. 3i) are rare, and the distribution of CTTs peak at slight supercoolings between 0 and -4 °C. The precipitating fraction as a function CTT has a notable peak ∼ -15 °C, which may be due to this temperature range

295 being the peak of vapor depositional growth rates (e.g., Fukuta and Takahashi, 1999; Wallace and Hobbs, 2006) increasing the likelihood of radar detectability, as also seen in Silber et al. (2021).

Alexander and Protat (2018) quantified the fraction of supercooled liquid water clouds at Cape Grim, Tasmania (40.7°S, 144.7°E) with ice virga below LCB using a ground-based lidar. They found that for stratocumulus layers with CTTs < -15 °, the fraction of precipitating ice virga clouds was ∼ 70-80%, but this fraction decreased to < 20% for CTTs warmer than -15

300 °C. Radenz et al. (2021) found a similarly small percentage of ice virga clouds for CTTs warmer than -15 °C using a radar-lidar approach over Punta Arenas, Chile (53.1 ° S, 70.9 °W). However, both of these studies limited their datasets to relatively optically and geometrically thin stratocumulus clouds. Here, the larger precipitating fraction at relatively warm supercooled CTTs (> -15 °C) may be due to the inclusion of optically and geometrically thicker layers (e.g., cumulus), particularly partially supercooled layers that precipitate in the liquid phase, are generally thicker, and precipitate quite frequently (Figs. 2e and 3c).

305 Fig. 3 shows that thicker clouds and those with colder CTTs are more likely to precipitate, but the cloud thickness and CTT are highly correlated since thicker clouds have higher CTHs and thus colder CTTs. To discriminate between these two cloud properties, the cloud-base precipitating fraction is projected onto CTT and cloud thickness by means of joint histograms in Fig. 4. As expected, the distribution shows that cloud thickness generally increases with decreasing CTT. However, the precipitating fraction generally increases for colder CTTs for the same cloud thickness, indicating that supercooled cloud

310 layers more readily precipitate than warm clouds (e.g., Mitchell et al., 1989; Senior and Mitchell, 1993; Tsushima et al., 2006; Hoose et al., 2008; Mülmenstädt et al., 2021). A stricter $Z_e$ threshold of -20 dBZ (Fig. 4b and as implied by Mace and Protat (2018a) to indicate light precipitation) shows this more clearly, where the precipitating fraction increases by up to a factor of 2 between CTTs of 0 and -15 °C for even relatively thin clouds (< ∼ 500 m). The exception to this is for cloud thicknesses < 200 m, where the majority of clouds do not precipitate regardless of their CTT.

### 315 3.2 Cloud-base Precipitation Rates ($R_{cb}$)

In total, 69% of identified cloud layers with CBHs > 250 m are precipitating from LCB. Of all precipitating cloud layers, ∼ 54% are supercooled, 22% are partially supercooled, and 24% are warm (legend of Fig. 5a). Precipitation rates are derived as described in Sect. 2.2.3 and the probability distribution is shown in Fig. 5a. The $R_{cb}$ distribution for all cloud layers peaks just under $10^{-1}$ mm hr$^{-1}$, where supercooled layers largely control the total $R_{cb}$ distribution. Warm cloud layers produce the



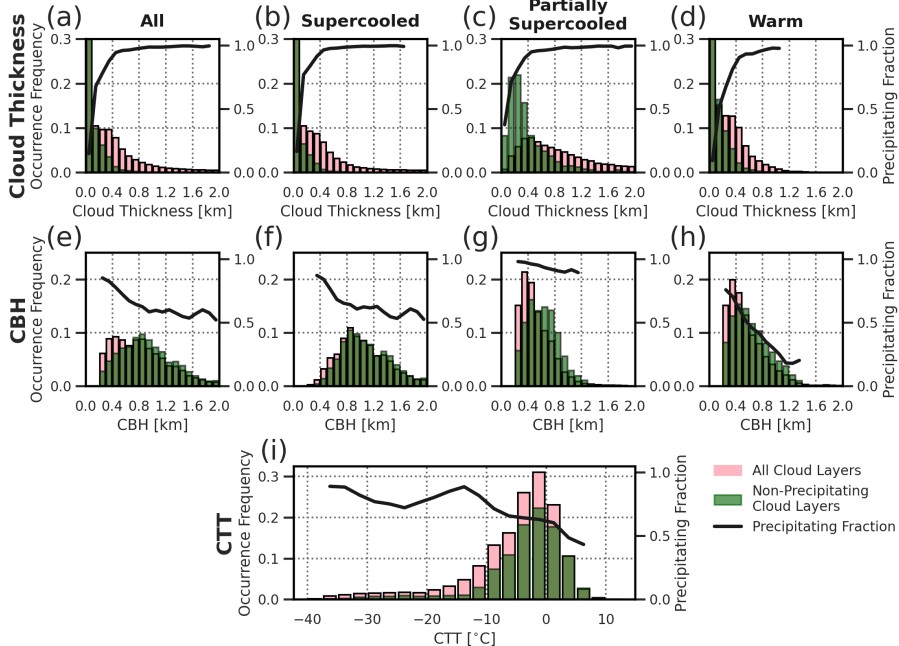

**Figure 3.** Occurrence frequency distributions of cloud thickness (top row), CBH (middle row), and CTT (bottom row) for all cloud layers (first column), supercooled layers (second column), partially supercooled layers (third column), and warm layers (last column). All cloud layers are shown as pink bars while non-precipitating cloud layers are shown as green bars. The precipitating fraction as a function of each cloud property bin is shown as a black line.

weakest $R_{cb}$ and peak between rates of $10^{-4}$ and $10^{-3}$ mm hr$^{-1}$. The partially supercooled $R_{cb}$ distribution is the narrowest with a peak just above $10^{-1}$ mm hr$^{-1}$. Both supercooled and partially supercooled $R_{cb}$ distributions are negatively skewed while warm cloud layer $R_{cb}$ distributions are positively skewed.

$R_{cb}$ distributions are further partitioned by CTT (Fig. 5b) and cloud thickness (Fig. 5c). $R_{cb}$ peak probabilities increase with decreasing CTT and increasing cloud thickness. $R_{cb}$ was also found to increase for decreasing CTT while controlling for cloud thickness (not shown), implying that colder clouds, regardless of their thickness, have higher $R_{cb}$, likely owing to the presence of ice precipitation.

### 3.3 Evaporation/Sublimation Below Cloud Base

Evaporation (or sublimation) below cloud base is evaluated in terms of the evaporated fraction, which is the fraction of layers with detectable cloud-base precipitation that is not continuous down to $h_{min}$. The evaporated fraction is shown as a function of CTT and cloud thickness via a joint histogram in Fig. 6a and as a function of surface RH (RH$_{sfc}$) and CBH in Fig. 6b. Evaporated fraction decreases with increasing cloud thickness. Thicker cloud layers are likely to have more vertically integrated condensate and have higher $R_{cb}$ such that thicker layers are more resilient to complete desiccation (Fig. 5c). Unsurprisingly, evaporated





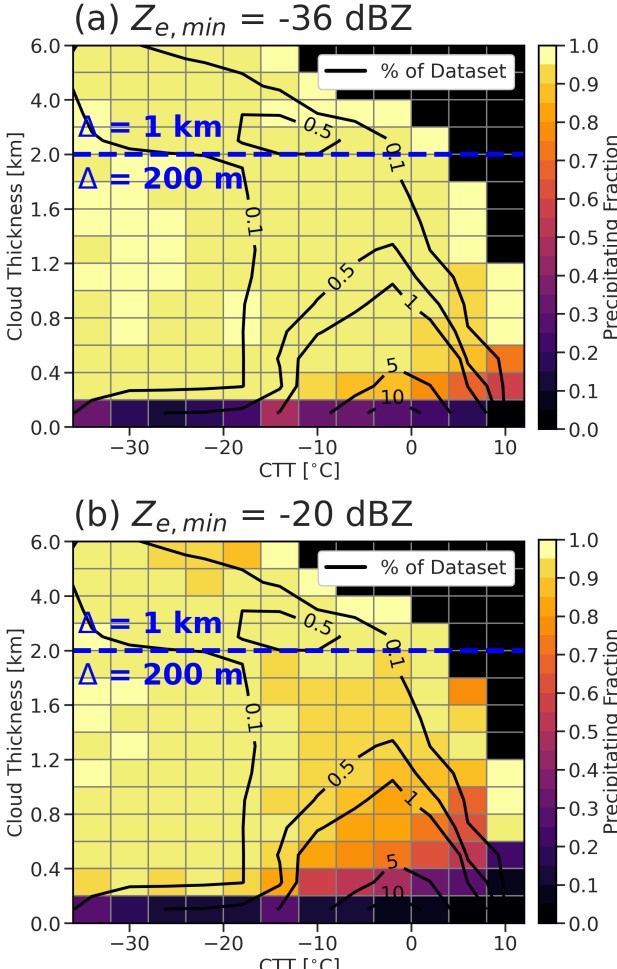

**Figure 4.** Joint histogram of CTT (abscissa) and cloud thickness (ordinate) shown as the percentage of the dataset in black contours and color-filled with the precipitating fraction for all samples within a given CTT-cloud thickness bin. Panel (a) uses $Z_{e,min}$ = -36 dBZ for detecting precipitating layers and panel (b) uses $Z_{e,min}$ = -20 dBZ. The bin width ($\Delta$) for CTT is 4 °C. For cloud thickness, $\Delta$ is split between two ranges. For thicknesses < 2 km, $\Delta$ = 200 m, while $\Delta$ = 1 km for thicknesses > 2 km, denoted by the horizontal dashed blue line.

fraction increases for decreasing CTT owing to the Clausius-Clapeyron relationship. This suggests that precipitation from supercooled cloud layers is more likely to evaporate/sublimate below LCB than precipitation from warm layers. This trend is
consistent with the larger decrease in supercooled precipitation occurrence at the surface relative to cloud base in supercooled layers compared to warm layers (Fig. 2c,d and g,h). In Fig. 6b, surface RH and CBH are expectedly correlated. The evaporated fraction increases for increasing CBH and decreasing RH, as cloud bases at higher altitudes have a larger depth of sub-cloud air for evaporation to act and are likely to be colder (barring temperature inversions).



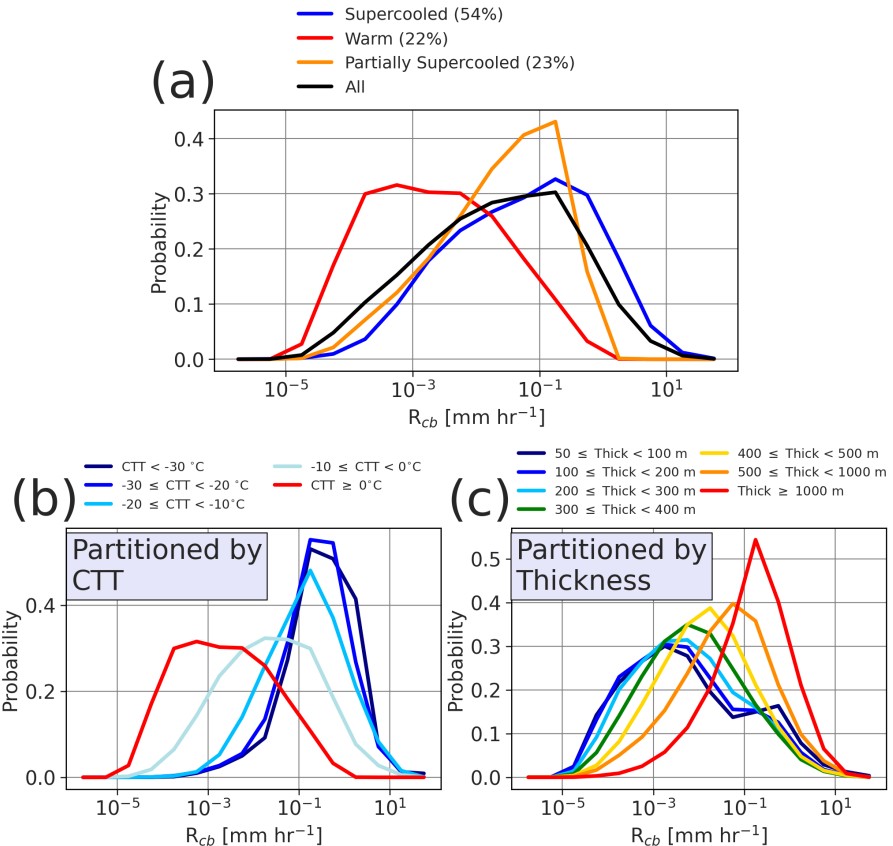

**Figure 5.** Probability distributions of $R_{cb}$ partitioned by (a) warm, partially supercooled, and supercooled layers, (b) CTT, and (c) cloud geometric thickness. In (a), the combined PDF of all layers is shown in black.

## 3.4 Special Cases

### 3.4.1 Optically Thin Cloud Layers

Cloud detection herein relies on the merged ceilometer dataset (CEIL), the CBHs for which are derived by the vendor's proprietary algorithm. Precipitation detection requires that reflectivity be coincident in the bin identified by CEIL, but a large proportion (27%) of clouds with CBHs > 250 m were optically thin where the CEIL-identified cloud base bins do not have coincident reflectivity. This is illustrated in Fig. 1, for example between $\sim$ 2335 UTC and 2350 UTC, where the ARM ceilometer's attenuated backscatter ($\beta_{\text{att}}$) observes values > $10^{-4}$ m$^{-1}$ sr$^{-1}$ (indicative of liquid cloud bases, Fig. 1f) but radar reflectivity during this time period (Fig. 1d) does not reach BASTA's $Z_{e,\text{min}}$ at that altitude. These layers are referred to as CEIL-only clouds.



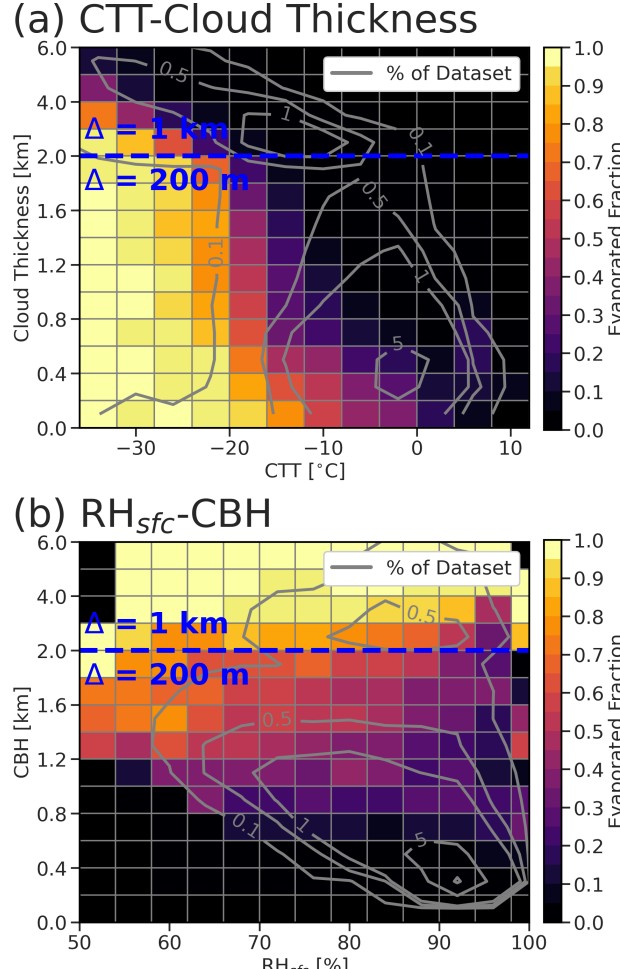

**Figure 6.** Joint histograms of (a) CTT and cloud thickness and (b) $RH_{sfc}$ and CBH with percentage of the dataset contoured in grey and the color-fill is evaporated fraction. The bin width ($\Delta$) for $RH_{sfc}$ is 5%. Bin widths for CBH and cloud thickness are split between two ranges. For thicknesses < 2 km or CBHs < 2 km AGL, $\Delta$ = 200 m, while $\Delta$ = 1 km for thicknesses > 2 km or CBHs > 2 km AGL, denoted by the horizontal dashed blue line.

Fig. 7 shows a scatterplot between the CBH and CBT for these CEIL-only cloud bases. The color-fill of each point on the scatterplot is the observation density and a histogram is shown on each axis for the one-dimensional observation density for
CBH and CBT ignoring the other variable. The majority of these optically thin clouds have bases < 2 km AGL (peaking ∼ 1 km AGL) and temperatures ranging from -10 °C and 5 °C. The median CBT for these clouds is -3 °C, indicating that many of these clouds are only very slightly supercooled.

Mace and Protat (2018a) determined that approximately 30% of clouds during the SO CAPRICORN I voyage were detected only by a lidar with no coincident layer-averaged reflectivity (as opposed to just considering reflectivity at cloud base as is





done here). Here, 20% of CEIL-only layers do obtain radar reflectivities above the noise floor within 100 m above LCB (not shown). Together, these indicate that 20–30% of clouds from MICRE and CAPRICORN I are representative of optically thin liquid layers unregistered by BASTA. We note that these layers were also evaluated during times with colocated soundings, in which sounding $RH_{liq}$ values often showed a high peak ($> 95\%$) at the same level with enhanced $\beta_{att}$ values where LCB is detected without coincident reflectivity (not shown). Their structure is often persistent with little vertical variability in the

LCB height and in some instances hydrometeors grow large enough to be intermittently detected by BASTA (for example in Fig. 1). Accounting for these optically thin clouds has important implications for defining $P_{cb}$ since these non-precipitating cloud layers are a non-negligible fraction of the normalizing cloud population. Because many studies have required that a cloud layer have coincident reflectivity (e.g., Lamer et al., 2020b; Silber et al., 2021), it is therefore possible that $P_{cb}$ for warm cloud layers is overestimated in such studies due to elimination of these optically thin layers from the cloud population. However,

for supercooled layers in which ice-phase precipitation can be "detached" from cloud base as it grows below LCB via vapor diffusion, $P_{cb}$ may still be underestimated (e.g., Silber et al., 2021). The prevalence of this cloud regime in other geographical regions is unclear, though Mace and Protat (2018a) also found this optically thin cloud type in $\sim 20\%$ of cloud layers over the ARM ENA site at Graciosa Island in the Azores (39 °N and 28 °W).

### 3.4.2 Near-surface Clouds and Fog

$P_{cb}$ calculations require the minimum CBH to be 250 m using $D_{min} = 100$ m. Of all CEIL-identified layers, 26% of cloud bases are $< 250$ m, which collectively are called "near-surface clouds". The $\beta_{att}$ profiles for these periods show repeating patterns of specific cloud morphology. Two case studies for these morphologies are discussed in Appendix F. In particular, Figs. F1 and F2 show CBHs identified below 150 m (within the BASTA "blind zone") and the $\beta_{att}$ profiles from CEIL show values $> 10^{-4}$ $m^{-1}$ $sr^{-1}$ at cloud base but with no significant reduction in $\beta_{att}$ below cloud base towards the surface. We consider these cases

to be fog, noting that this is a broad definition that may include deliquescenced aerosols that produce haze.

A simple fog identification algorithm was developed to identify cases where the cloud base $\beta_{att} \geq 10^{-4.5}$ $m^{-1}$ $sr^{-1}$ and does not decrease by at least an order of magnitude below cloud base. Profiles matching these specifications occurred 15% of the time (accounting for 57% of near-surface clouds). We examine distributions of surface measurements for all near-surface clouds and for those identified as fog in Fig. 8. $RH_{sfc}$ values exceed 90% for almost the entirety of the distributions for near-

surface clouds and fog, with some tendency for smaller values for non-fog profiles, supporting the possibility of haze in some instances. Surface temperatures are always above freezing during this time period, peaking around 7 °C with no significant differences between the distributions of near-surface clouds and fog. Surface wind speeds also show no significant differences for fog relative to all near-surface clouds, but we note the persistence of rather strong surface wind speeds (distribution modes $\sim 20$ m s$^{-1}$), indicating that these fog events are likely of the advective type rather than radiative fog, which requires calm

surface conditions. The fog formation processes may be analogous to those during Arctic air formation (Tjernström et al., 2019). Mace and Protat (2018a) reported that the air temperature was colder than the sea surface temperature except for a few days during the SO CAPRICORN I voyage spanning 43 to 53 °S, equatorward of Macquarie Island, which may explain the lack of similarly abundant near-surface clouds reported in their study. While other recent SO voyages reached the edge of



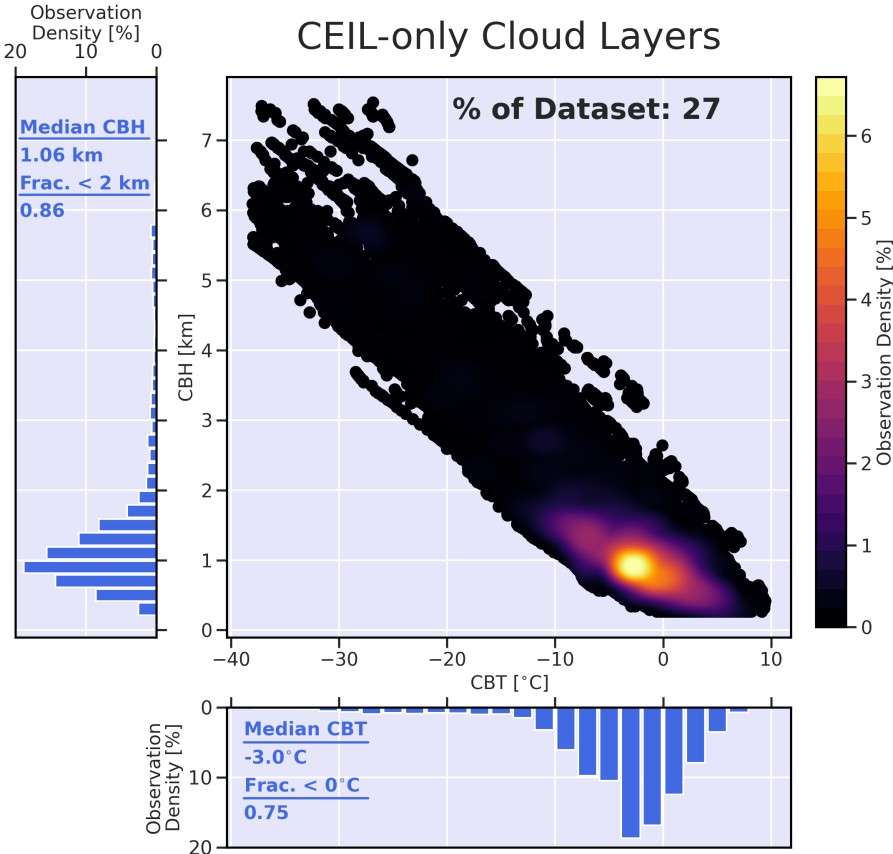

**Figure 7.** Scatterplot of the 27% of cloud bases above 250 m AGL where a cloud base is detected only by CEIL (i.e., no coincident radar reflectivity) as a function of CBT (abscissa) and CBH (ordinate). Points are color-filled with the observation density. One dimensional observation density histograms are also plotted on the respective axes.

Antarctica (e.g., Kremser et al., 2021; McFarquhar et al., 2021) , none occurred during the coldest months of the year and each

was relatively short compared to the annual cycle observed during MICRE. Indeed, fog detections during MICRE were more frequent in Austral winter and transition months than during Austral summer (not shown).

Even though CBH is too low to establish precipitation below it, valid radar reflectivity was identified between 150 and 250 m in $> 98\%$ of all near-surface clouds and fog layers. Valid radar reflectivity values close to the surface are converted to precipitation rates using the Comstock et al. (2004) $Z$-$R$ relationship. The bottom row of Fig. 8 shows the layer-averaged $Z_e$ between

150 and 250m AGL ($\overline{Z_{e,150-250m}}$) and the surface precipitation rate ($R_{\mathrm{sfc}}$) derived from radar reflectivity. Distributions of $\overline{Z_{e,150-250m}}$ are largely similar between near-surface clouds and fog, although fog layers are shifted slightly toward larger values and thus slightly larger $R_{\mathrm{sfc}}$. Note that $> 90\%$ of the distributions have $R_{\mathrm{sfc}} > 10^{-3}$ mm hr$^{-1}$, suggesting non-negligible contributions to $R_{\mathrm{sfc}}$ by these fog layers and near-surface clouds.



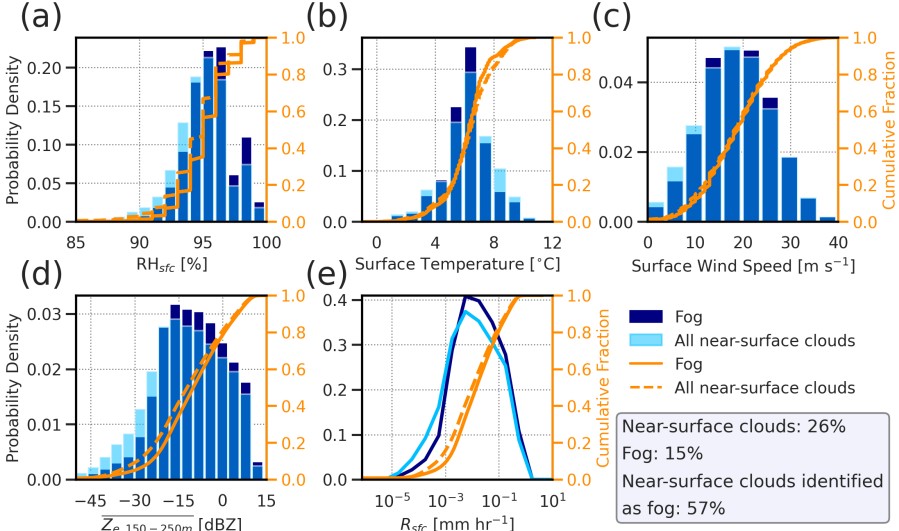

**Figure 8.** Probability distributions of (a) RH$_{sfc}$, (b) surface temperature, (c) surface wind speed, (d) layer-averaged $Z_e$ between 150 and 250 m AGL ($\overline{Z_{e,150-250m}}$), and (e) the retrieved precipitation rate from BASTA between 150 and 250 m AGL ($R_{sfc}$). Light blue bars are for all near-surface clouds and dark blue bars are for near-surface clouds identified as fog. The solid and dashed lines show the cumulative fraction for profiles identified as fog and for all near-surface clouds, respectively. The text box in the lower right shows the percentage of cloud profiles identified as near-surface clouds, the percentage of cloud profiles identified as fog, and the percentage of near-surface cloud profiles identified as fog.

The Arctic and Antarctic sites evaluated by Silber et al. (2021) required an $h_{min}$ of 300 m, such that near-surface clouds
(including potential fog) were not considered, but we note that fog features were seen to some degree in the Arctic data
from NSA. Because the radar "blind zone" (i.e., the surface through $h_{min}$) limits the detection of hydrometeors within this
range, it is routine for studies to truncate cloud detection from ground-based instrumentation to above $h_{min}$. However, the large
proportion of CBHs identified below 250 m (26% of all clouds) in this study implies the need for more robust quantification
of fog and near-surface clouds. Indeed, a 30-year climatology (1952-1981) of global cloud type distributions from ship-based
observations showed a global peak in fog frequency of occurrence between a latitudinal band from 40 to 70 °S, including over
Macquarie Island's longitude (Warren et al., 1988). They showed a seasonal cycle that appears to maximize during Austral
summer, suggesting that fog formation mechanisms are not limited to Arctic air formation during Austral winter discussed
above. In addition, Kuma et al. (2020) used ship-based ceilometer data from multiple SO voyages and found that occurrence
frequencies of CBHs peak below 500 m AGL and often very near the surface, indicative of fog, and that these low clouds were
often associated with near-surface air temperatures $< 0$ °C and warmer than the SST, analogous to Arctic air formation.



## 4 ESM Evaluation

### 4.1 Model Setup

We next demonstrate use of the merged MICRE dataset to evaluate a 9-year (2012-2020), global free-running (i.e., no nudging) simulation using the NASA GISS-ModelE3 ESM. In brief, the simulation used here employs 2 x 2.5 ° resolution and 110
vertical levels. The model configuration is the same as used by Cesana et al. (2021), also referred to as GISS-ModelE3-Phys in that study's supporting information, denoting a configuration that uses the default set of tuning parameters and an alternative entrainment closure for moist convection. Other aspects of the model are summarized by Cesana et al. (2021) and references therein. The simulation is initialized on November 1, 2011 for two months of model spin-up and prescribes sea surface temperatures using a climatology following the Atmospheric Model Intercomparison Project (AMIP) specifications (Gates, 1992;
Gates et al., 1999). Aerosol profiles are prescribed as a single-mode log-normal size distribution with regionally and seasonally varying concentrations and activation follows from Abdul-Razzak et al. (1998). For stratiform cloud microphysics, a modified version of the Gettelman and Morrison (2015) two-moment bulk microphysics scheme (MG2) is used that includes prognostic precipitation. Convective cloud microphysics are described in Cesana et al. (2019b). Both the stratiform and convective schemes include the following four hydrometeor classes: cloud liquid water, cloud ice, precipitating liquid water, and
precipitating ice.

### 4.2 EMC$^2$ Instrument Simulator Application

For application of EMC$^2$, microphysical variables required for the simulation of radar and lidar moments are output in the grid cell containing Macquarie Island at model physics time-step frequency (30 minutes) as instantaneous values for comparison with observations. EMC$^2$ offers two approaches for remote sensing calculations, including a radiation scheme logic that gen-
eralizes hydrometeor fractions and uses bulk scattering calculations for specific size distributions, and a microphysics logic that uses single-particle scattering calculations with the model's predicted particle size distributions. Here, the microphysics scheme logic is used. After providing to EMC$^2$ a user-defined number of subcolumns (taken here as 8), hydrometeors are allocated to the subcolumns by translating the volume fraction of the model's hydrometeor class to a number of hydrometeor-containing and hydrometeor-free subcolumn bins. The maximum-random overlap approach (Tian and Curry, 1989; Fan et al.,
2011; Hillman et al., 2018) is then applied from the top down, which preferentially extends cloud layers vertically within a subcolumn and retains vertical continuity of cloud and precipitation features. Further details of subcolumn generation and forward simulation can be found in Silber et al. (2022).

A 24-hr example of variables simulated by EMC$^2$ is shown in Fig. 9 for a slightly supercooled, precipitating stratocumulus case. Three of the eight subcolumns are used to demonstrate simulated 95 GHz attenuated $Z_e$, 910 nm $\beta_{att}$, and GISS-ModelE3
precipitation rates. Precipitation detection for GISS-ModelE3 is performed in a similar manner as for MICRE observations with a few differences. Rather than performing a CBH identification algorithm via the simulated 910 nm ceilometer $\beta_{att}$, LCB is identified explicitly as the lowest altitude subcolumn pixel in time-height space that contains cloud liquid water content (CLWC). This treatment implies an LCB for every column that contributes to total liquid cloud fraction. We note that LCB



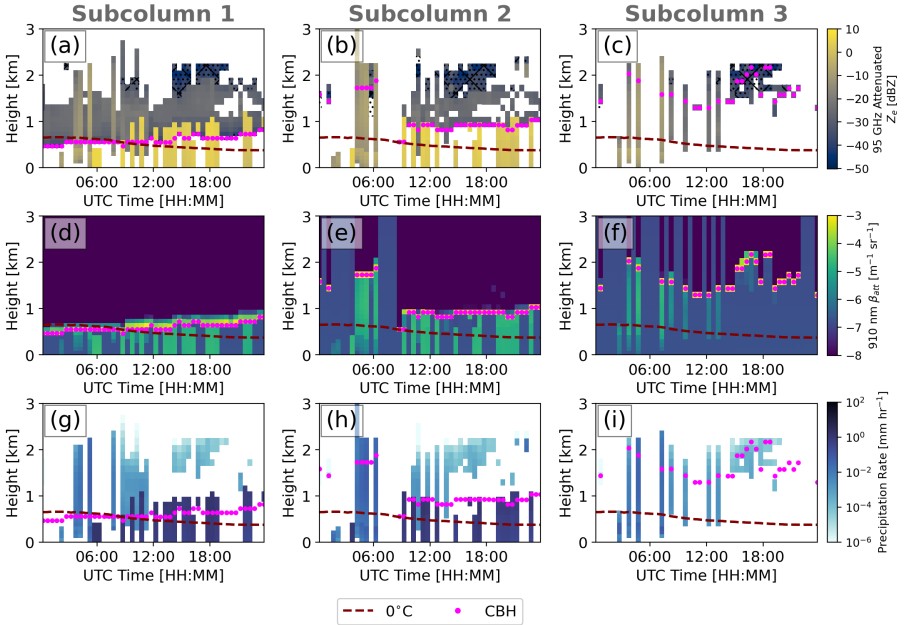

**Figure 9.** Example 24-hr time-height series of EMC$^2$ simulated (a)-(c) 95 GHz attenuated $Z_e$, (d)-(e) 910 nm $\beta_{att}$, and (g)-(i) GISS-ModelE3 precipitation rate (sum of convective and stratiform precipitation rates) for a slightly supercooled, precipitating stratocumulus case. The three columns represent three out of eight subcolumns generated using EMC$^2$. CBH is denoted by magenta dots and the 0°C isotherm is shown by a red dashed line. Hatching in (a)-(c) represents hydrometeor-containing grid cells with reflectivity lower than the BASTA $Z_{e,\min}$.

identified with this method is most often colocated with locally enhanced simulated $\beta_{att} > 10^{-4}$ m$^{-1}$ sr$^{-1}$ (Fig. 9d-f). For
comparison with the observational approach, we find that the cloud occurrence frequency is not sensitive to CLWC or $\beta_{att}$ thresholding beyond an arbitrary value that is indicative of non-negligible liquid cloud mass (see Appendix G and Fig. G1).

Precipitation detection is then performed at the same pixel as LCB. While the GISS-ModelE3 convective and stratiform precipitation schemes inform whether or not the precipitation process is active immediately at cloud base, precipitation is only considered detectable for comparison with MICRE observations where the simulated 95 GHz attenuated $Z_e$ is above the
BASTA noise floor. If a column pixel has a $Z_e$ value above the noise floor coincident with hydrometeor mass from a precipitating hydrometeor species at LCB, the cloud layer is diagnosed as precipitating. The explicit mass-weighted precipitation rate from the model at that pixel is then taken as the cloud-base precipitation rate (i.e., $R_{cb}$). We note that $P_{cb}$ is not sensitive to an arbitrary minimum $R_{cb}$ threshold (Appendix G). Finally, all algorithm limits applied to the MICRE dataset are applied to the GISS-ModelE3 simulation. Namely, LCBs are limited to altitudes below 7.7 km AGL, CBTs and CTTs are limited to warmer
than -38 °C, and noise floor restrictions from 95 GHz attenuated $Z_e$ emulating BASTA are applied to cloud and precipitation retrievals.





**Table 1.** Comparison of cloud and precipitation properties between the MICRE dataset and the 9-year GISS-ModelE3 ESM simulation. Indentations are used to represent percentages relative to the normalizing population given one indentation level above, where the top-level normalizing population for MICRE is $\sim$ 1 year of valid profiling instrument data and for GISS-ModelE3 is the 9 years of simulation data. Values in parentheses under the GISS-ModelE3 columns are absolute differences from MICRE observations.

| | All Layers | | Supercooled | | Partially Supercooled | | Warm | |
|---|---|---|---|---|---|---|---|---|
| | MICRE | E3 | MICRE | E3 | MICRE | E3 | MICRE | E3 |
| Total Cloud Occurrence Frequency (%) | 76 | 57 (-19) | - | - | - | - | - | - |
| CBH < 250 m (%) | 26 | 26 (0.0) | - | - | - | - | - | - |
| Fog (%) | 57 | 69 (+12) | - | - | - | - | - | - |
| CBH > 250 m (%) | 74 | 74 (0.0) | 61 | 78 (+17) | - | - | 39 | 23 (-16) |
| CEIL-only (%) | 27 | 31 (+4.0) | 75 | 87 (+12) | - | - | 25 | 13 (-12) |
| $P_{\text{cb}}$ (%) | 69 | 55 (-14) | 63 | 50 (-13) | 97 | 93 (-4.0) | 65 | 53 (-12) |
| Evaporated Fraction (%) | 38 | 53 (+15) | 49 | 57 (+8.0) | 12 | 26 (+14) | 36 | 71 (+35) |
| Supercooled Partitioning (%) | - | - | 54 | 70 (+16) | 24 | 19 (-5.0) | 22 | 11 (-11) |
| $P_{\text{sfc}}$ (%) | 54 | 29 (-25) | 45 | 24 (-21) | 90 | 72 (-18) | 53 | 18 (-35) |
| Total Fog Occurrence Frequency (%) | 15 | 18 (+3.0) | - | - | - | - | - | - |

## 4.3 Comparison with MICRE

Table 1 provides a comparison of cloud and precipitation properties between MICRE and the GISS-ModelE3 simulation. All values are percentages relative to a normalizing population, given as the population one indentation level above. The top-level normalizing population for MICRE is $\sim$ 1 year of operational vertical profiles passing quality control, while the GISS-ModelE3 top-level normalizing population is 9 years of simulated profiles. Absolute differences between MICRE and GISS-ModelE3 statistics are denoted in parentheses. GISS-ModelE3 produces a total cloud occurrence frequency of 57%, which is 19 percentage points lower than the MICRE observations. Of all cloudy profiles, 74 % of GISS-ModelE3 CBHs are higher than 250 m, which is the same percentage as MICRE. Supercooled layers account for 78% of all CBHs > 250 m AGL in GISS-ModelE3 and 61% in MICRE. For CBHs > 250 m AGL, 31% of cloud bases in GISS-ModelE3 did not have coincident $Z_e$ above the noise floor compared to 27% of MICRE cloud bases being identified only by CEIL.

For CBHs > 250 m, 55% are precipitating from LCB in GISS-ModelE3 compared to 69% in MICRE. $P_{\text{cb}}$ as a function of $Z_{e,\text{min}}$ is shown in Fig. 10 for GISS-ModelE3 and MICRE for all cloud layers and partitioned by supercooled, partially





supercooled, and warm layers. This $P_{cb}$ projection illustrates both the radar sensitivity and the contribution to $P_{cb}$ by cloud
bases precipitating at a given $Z_e$ threshold. All layers precipitate less frequently in GISS-ModelE3 compared to MICRE,
which is constant regardless of $Z_{e,\min}$. Partially supercooled cloud layers precipitate most frequently in GISS-ModelE3, with
only a decrease by 4 percentage points relative to MICRE, while supercooled and warm layers precipitate less frequently in
GISS-ModelE3 by 14 and 12 percentage, respectively, at the BASTA sensitivity.

For supercooled and partially supercooled cloud layers, $P_{cb}$ is relatively insensitive to $Z_{e,\min} <$ -36 dBZ (region to the left
of the light blue dashed line in Fig. 10), which occupies the lowest 1 km AGL of BASTA's range. However, warm cloud
layers populate this $Z_e$ range since warm CBH is generally $<$ 1 km (see Fig. 3h). This $Z_e$ range accounts for a 10% increase
in warm-layer $P_{cb}$ in GISS-ModelE3 and a 15% increase in MICRE when decreasing $Z_{e,\min}$ from -36 dBZ to -50 dBZ. We
emphasize that in both MICRE and GISS-ModelE3, although the $P_{cb}$ for supercooled and warm layers listed in Table 1 are
similar, a large portion of warm-layer $P_{cb}$ is attributable to cloud layers with sub-cloud base $Z_e <$ -36 dBZ. At higher $Z_{e,\min}$
thresholds (e.g., $>$ -36 dBZ), supercooled cloud layers consistently precipitate more frequently than warm layers. Overall,
GISS-ModelE3 produces a systematic low bias in $P_{cb}$ relative to MICRE regardless of the cloud top temperature or $Z_{e,\min}$
threshold. One potential cause for lower $P_{cb}$ in GISS-ModelE3 is the lack of interactive aerosol, which is prescribed in the
current runs and should be investigated in future studies.

Precipitating layers also evaporate more frequently in GISS-ModelE3 compared to MICRE. The evaporated fraction is 38%
in MICRE, but 53% in GISS-ModelE3. All levels of supercooling produce excessive evaporated fractions, but the largest bias
occurs in warm clouds, where the evaporated fraction is 71% in GISS-ModelE3 compared to 36% in MICRE. This excessive
evaporation results in a $P_{sfc}$ of only 18% in GISS-ModelE3 relative to 53% in MICRE.

Distributions of GISS-ModelE3 $R_{cb}$ are shown in Fig. 11 and separated by supercooling, CTT, and cloud thickness, as in
Fig. 5. The MICRE $R_{cb}$ PDF is also shown in Fig. 11a. GISS-ModelE3 captures trends in $R_{cb}$ that are present in the MICRE
observations well, whereby supercooled layers have higher $R_{cb}$ relative to warm layers and partially supercooled layers have
the highest $R_{cb}$. Precipitation rates also increase with colder CTT and with larger cloud geometric thickness, as was seen
in the MICRE dataset (Fig. 5b-c). The total $R_{cb}$ distribution for both MICRE and GISS-ModelE3 are largely controlled by
supercooled cloud layers, which account for 70% of the distribution in GISS-ModelE3 compared to 54% in MICRE (Table 1).

Finally, the same fog identification algorithm applied to the MICRE dataset in Sect. 3.4.2 is applied here. Fog is identified
18% of the time in GISS-ModelE3 compared to 15% in MICRE, accounting for 69% and 57% of near-surface cloud layers
in GISS-ModelE3 and MICRE, respectively. This near agreement indicates that these near-surface cloud layers commonly
observed during MICRE are to some degree represented in GISS-ModelE3.

## 5  Discussion

### 5.1  Implications for ESMs

MICRE provides a unique year-long dataset for observing cloud and precipitation properties over the remote SO. A common
shortcoming of CMIP5 ESMs over the SO is a lack of clouds in general that results in excessive absorbed shortwave radiation




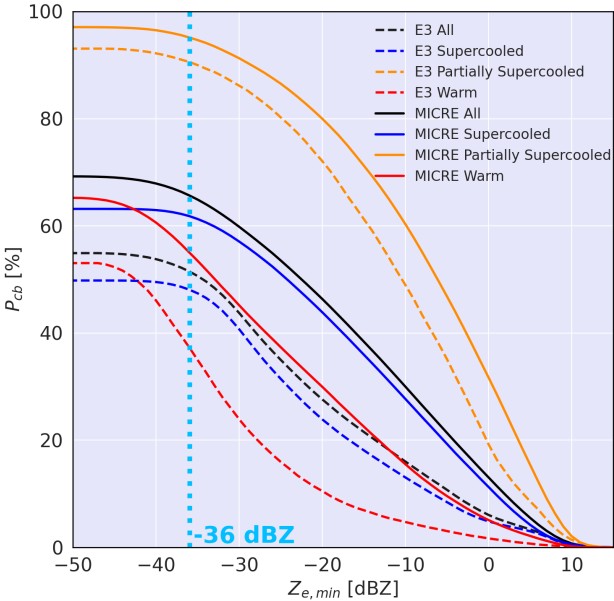

**Figure 10.** Cloud base precipitation occurrence frequency ($P_{cb}$) as a function of $Z_{e,\mathrm{min}}$ for the GISS-ModelE3 simulation (dashed lines) and for MICRE (solid lines), showing all cloud layers in black, supercooled layers in blue, partially supercooled layers in orange, and warm layers in red.

at the surface relative to observations (e.g., Trenberth and Fasullo, 2010; Bodas-Salcedo et al., 2012, 2014; Flato et al., 2013; Cesana et al., 2022). Conversely, some CMIP6 models improved this bias and now simulate too much stratocumulus that are not reflective enough (e.g., Schuddeboom and McDonald, 2021). In the current study, the occurrence frequency of liquid-based

clouds is 57% in GISS-ModelE3 compared to 76% in MICRE (with month-to-month variability of $\sim 10$ percentage points). The majority of LCBs in MICRE and in GISS-ModelE3 are supercooled, which is consistent with space-borne documentation of ubiquitous supercooled low-level liquid clouds (e.g. Morrison et al., 2011; Huang et al., 2012; Cesana and Chepfer, 2013; Chubb et al., 2013; Bodas-Salcedo et al., 2016). Even though GISS-ModelE3 produces fewer liquid-based clouds relative to observations, the majority of these clouds are indeed supercooled. Kay et al. (2016a) found that the Community Earth

System Model (CESM1; Hurrell et al., 2013) with the Community Atmosphere Model (CAM5) produced too few persistent supercooled liquid cloud layers and too much ice over the SO relative to satellite observations due to a preferential glaciation of simulated supercooled clouds. However, we note that here the supercooled $P_{cb}$ in GISS-ModelE3 is weaker than observed, suggesting that a lack of simulated supercooled cloud in GISS-ModelE3 may not be caused by a tendency for supercooled liquid clouds to glaciate and precipitate quickly.

The finding that supercooled cloud layers precipitate more readily than warm cloud layers for the same geometric thickness has implications for precipitation behavior in a warming climate. Mülmenstädt et al. (2021) discuss a negative cloud radiative feedback (i.e., cooling effect) in which a shift from ice and mixed-phase clouds to mostly liquid clouds in a warming climate



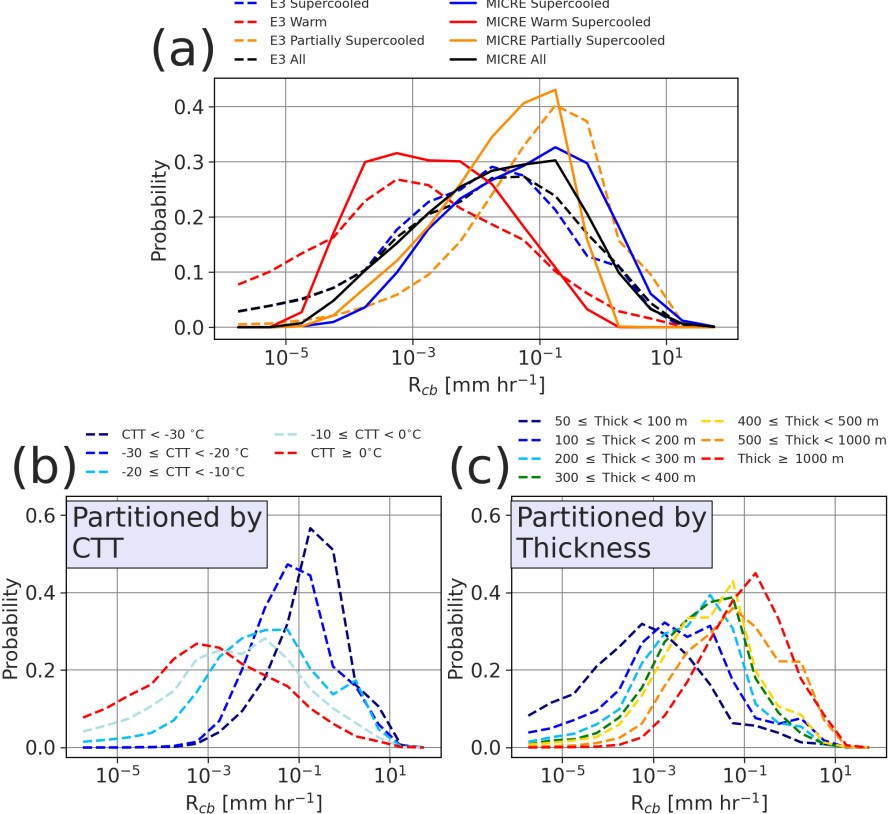

**Figure 11.** Probability distributions of GISS-ModelE3 $R_{cb}$ (dashed lines) partitioned by (a) warm, partially supercooled, and supercooled layers, (b) CTT, and (c) cloud geometric thickness. In (a), the combined PDF of all layers is shown in black and the MICRE $R_{cb}$ PDFs are shown as solid lines.

leads to more reflective clouds (optical feedback component) with a longer desiccation timescale (described as a so-called "lifetime" feedback component, where "lifetime" metaphorically refers to an increase in the horizontal extent and residence

time of cloud condensate in the atmosphere). However, this negative cloud feedback is modulated by how readily warm clouds precipitate. Studies that compare ESM warm-rain precipitation probability to space-borne active remote sensors show a relatively ubiquitous bias in which warm clouds precipitate too readily (e.g. Stephens et al., 2010; Suzuki et al., 2015; Jing et al., 2017; Kay et al., 2018). Indeed, Mülmenstädt et al. (2021) found in ESM simulations that a 4-K increase in surface temperature led to an increase in warm rain fraction over the SO, increasing the optical feedback component. However, they found

that warm-rain precipitation efficiency was high-biased relative to satellite observations, thereby reducing the efficiency of the lifetime feedback component. By reducing the warm-rain probability in the ESM to better agree with satellite observations, they found that the lifetime feedback component was three times larger than that in the default model owing to an increase in liquid water path.





Here, we find that warm clouds precipitate *less* frequently in GISS-ModelE3 relative to ground-based observations, which is
inconsistent with literature consensus based on satellite observations. Such differing conclusions could arise if GISS-ModelE3
behaves differently than other ESMs, if satellite observations underestimate precipitation occurrence frequency relative to colo-
cated ground-based observations, or if the model-observation comparisons consider substantially different conditions owing
to sampling or methodology. Fig. 2h showed $P_{cb}$ for all liquid-based clouds using the sensitivity and vertical resolution of
BASTA and for CloudSat 2C-PC "certain" and "possible" products, where $P_{cb}$ decreased from 70% for BASTA to 35%
("possible") and 20% ("certain") for 2C-PC. Although the sensitivity and vertical resolution of CloudSat suggested by Fig. 2h
does not account for CloudSat's data characteristics below 750 m AGL, this is roughly consistent with Tansey et al. (2022, see
their Fig. 10), who showed that liquid-phase surface precipitation frequency decreased by 30% in their ground-based dataset
compared to CloudSat. This comparison also implies that the GISS-ModelE3 $P_{cb}$ of 55% could be larger than CloudSat sug-
gests, but confirming that would require applying EMC$^2$ to GISS-ModelE3 outputs with CloudSat rather than BASTA radar
characteristics, an exercise that would still leave open the questions of sampling and methodology that are not trivially dis-
missed. Reconciling these differing conclusions regarding ESM precipitation occurrence to which model results are sensitive
(Mülmenstädt et al., 2021) will motivate further work to robustly evaluate models simultaneously against both ground-based
observations and satellite observations, while directly comparing ground-based and space-based observations as demonstrated
by Tansey et al. (2022). Additionally, ESM evaluation methodology using ground-based versus space-based simulators is wor-
thy of further investigation since results and conclusions drawn can be sensitive to the representation of model physics (e.g.,
Cesana et al., 2021).

This study also found that GISS-ModelE3 precipitation evaporates too frequently before it reaches within 250 m of the
surface, which can be expected to influence the cloud condensate budget in a number of competing ways. For example, sub-
cloud evaporation can act as a condensate sink by stabilizing the boundary layer (decreasing vertical mixing and cloud amount),
but can also act as a source of moisture in turbulent regions, where the condensate is not entirely lost to the surface through
precipitation and therefore is a moisture source for condensation to later occur.

Although we do not seek to actively address the model biases presented herein, these findings stress the importance of
understanding cloud and precipitation properties from a process-oriented perspective and using a simulator approach to account
for both observational limitations and consistency with model physics. We leave further in-depth assessment of the model's
physical mechanisms responsible for model-observations differences for future work. Ideally, future analyses should evaluate
thermodynamic and cloud conditions simultaneously over multiple sites in order to more robustly establish process-based
mechanisms and link them to leading biases. Indeed, Fiddes et al. (2022) evaluated nudged simulations by the Australian
Community Climate and Earth System Simulator (ACCESS) atmosphere model against satellite observations over the SO and
found that even when cloud radiative biases were small on average, cloud properties such as cloud fraction and vertically
integrated condensate can remain large.



## 5.2 Related Studies

Tansey et al. (2022) analyzed the same year of MICRE data and found that surface precipitation occurs $44 \pm 4\%$ of the time during the campaign. In the current study, a cloud occurrence frequency of 76% and a $P_{\text{sfc}}$ of 54% (Table 1) implies a campaign-long surface precipitation occurrence frequency of $\sim 41\%$, indicating good agreement with their study. Tansey et al. (2022) found precipitation to be primarily composed of small particles ($< 1$ mm in diameter) and found a significant contribution from light rain rates ($< 0.5$ mm hr$^{-1}$) that accounted for 11% of accumulated surface precipitation. Similar contributions by light rain rates were documented by Wang et al. (2015).

Similar observational analyses have been performed at other geographic locations. For example, Silber et al. (2021) documented the $P_{\text{cb}}$ of supercooled liquid-bearing layers at an Antarctic Site (McMurdo Station, Antarctica) during the ARM West Antarctic Radiation Experiment (AWARE; Lubin et al., 2020a) and at an Arctic site (NSA). They used soundings with an RH$_{\text{liq}}$ threshold to identify cloud boundaries combined with the ARM Ka-band Zenith Radar (KAZR; Widener et al., 2012) at both polar sites to detect sub-cloud precipitation. They found that 85% (75%) of supercooled clouds were precipitating from LCB at the Arctic (Antarctic) site. McMurdo Station is located at 77.8 °S and 166.7 °E, roughly 22.5 ° south and 8° east of Macquarie Island. We note that KAZRs have sensitivities around -50 dBZ at 1 km AGL (compared to -36 dBZ for BASTA during MICRE), although their $h_{\text{min}}$ is typically higher (e.g., Silber et al., 2021). When considering only $Z_e > $ -36 dBZ (below which supercooled clouds in this study are insensitive, see Fig. 10), the $P_{\text{cb}}$ at McMurdo Station from supercooled cloud layers per Silber et al. (see 2021, their Fig. 1b) was $\sim 70\%$ while in MICRE it was $\sim 61\%$ (see Fig. 2). Different cloud morphologies exist between Macquarie Island and McMurdo station, even for supercooled layers, due to Macquarie Island's location north of the Polar front and potential effects of terrain at McMurdo Station. The 9% absolute difference in supercooled cloud $P_{\text{cb}}$ between the stations may also lie within their summed uncertainties owing to relatively short deployments for the purposes of a climatology.

Lamer et al. (2020b) used 3 years of data from the ARM ENA observatory to evaluate cloud and precipitation properties in post-cold frontal subsidence regimes using a ceilometer and a radar, also taking a similar approach. They found 80% of cloud layers in subsidence regimes to be precipitating. The higher $P_{\text{cb}}$ of 80% over ENA compared to MICRE may be due to the requirement that their cloud layers produce detectable reflectivity above lidar-identified cloud base, whereas here we also consider optically thin, non-precipitating clouds without coincident radar reflectivity above the noise floor, which increases the normalizing cloud population in our study. They also related cloud geometric thickness to $P_{\text{cb}}$ and $R_{\text{cb}}$ and found that $P_{\text{cb}}$ increases with increasing cloud geometric thickness, which is consistent with this study and results in Silber et al. (2021). $R_{\text{cb}}$ also increased with increasing cloud thickness in Lamer et al. (2020b), agreeing with our study and following from other observational studies suggesting that $R_{\text{cb}}$ scales with cloud thickness (e.g., Yang et al., 2018; vanZanten et al., 2005). Also similar to our study, Lamer et al. (2020b) found a higher likelihood for precipitation to reach the surface from deeper cloud layers.



### 5.3 Implications for Satellites

Silber et al. (2021) reconciled discrepancies between ground-based observations that indicate polar supercooled clouds as
nearly continuously precipitating lightly from LCB and much lower precipitation frequencies derived from space-borne instruments, based on differences in radar sensitivity and vertical resolution. Here we find a similar importance of radar sensitivity (Fig. 2) spanning clouds with both supercooled and warm CBTs, especially for $Z_e$ values that represent the weakest precipitation fluxes. Satellite observing platforms experience ground clutter near the surface and are thus unable to detect clouds within the lowest 0.75-1 km AGL. During MICRE, the majority of warm-based clouds and a large fraction of supercooled clouds have
CBHs < 1 km (Fig. 3). This high frequency of CBHs < 1 km suggests severe limitations for detection of cloud-base precipitation from current spaceborne instrumentation. Indeed, CloudSat's $Z_{e,\min}$ = -15 dBZ and vertical resolution ($D_{\min}$) = 250 m would yield a $P_{cb}$ = 40%, nearly 30 percentage points lower than shown here from BASTA (see Fig. 2a). As discussed by Silber et al. (2021), the future EarthCARE mission's Cloud Profiling Radar will be more sensitive and better at establishing light precipitation processes (Kollias et al., 2014; Illingworth et al., 2015). However, given that all current and future ground-based
and satellite instrument datasets will have limitations in terms of geographical and temporal coverage, instrument resolution and sensitivity, and factors such as attenuation and ground clutter, a simulator approach provides a robust pathway to enable fusion of spaceborne and ground-based platforms for reliable model evaluation, as pioneered by tools such as the 2nd version of the Cloud Feedback Model Intercomparison Project Observational Simulator Package (COSP2; Swales et al., 2018).

### 5.4 Caveats and Guidance for Future Southern Ocean Campaigns

Macquarie Island's latitude of 54.5 °S is not necessarily expected to be representative of the vast SO. For example, Fiddes et al. (2022) split the SO into three latitudinally bound regions and found that model biases in cloud phase and morphology were different among the three regions. Expansion of the results here to other latitudes should be focus of future work. In addition, we note that Tansey et al. (2022) documented that MICRE summer surface precipitation was anomalously high relative to a long-term tipping bucket record from Macquarie Island, indicating the need for more routine measurement platforms over the
SO and robust satellite supplementation in order to place the results of this study within the context of the broader Macquarie Island and SO climatologies.

    Finally, this study illustrates a number of needs for future ground-based missions over the SO. Longer deployments (order of years) are needed to robustly characterize the cloud climatology and seasonal variability over Macquarie Island. The cloud properties presented herein could be more robustly analyzed with higher-capability lidar instrumentation. Although polariza-
tion lidar capability was present during MICRE, it was not available for statistical evaluation. Verifying the phase of cloud base detections via polarization lidar is needed since this is difficult to determine through ceilometer attenuated backscatter alone, though Guyot et al. (2022) demonstrated a data-driven approach to classify cloud phase based on ceilometer attenuated backscatter gradients. We note also BASTA's sensitivity would have been higher during MICRE had the low noise amplifier been operational (see Appendix B). In particular, determining the height of LCB presented a leading uncertainty in this
study, including that associated with the proprietary vendor algorithm used to detect LCB height. Low cloud bases within the





radar "blind zone" should also be investigated further over the SO (e.g. Maahn et al., 2014; Kuma et al., 2020) . For example, Alexander and Protat (2018) found that $\sim$ 15% of lidar-identified cloud bases at Cape Grim, Tasmania (40.7°S, 144.7°E) were below 200 m AGL. During MICRE, about a quarter of ceilometer-identified CBHs were below 250 m. Over half of these surface-based clouds during MICRE were representative of fog, which with the exception of Kuma et al. (2020) has not been
extensively studied over the SO and also deserves further investigation.

## 6 Conclusions

This study evaluated cloud and precipitation properties using ground-based profiling instrumentation at the Southern Ocean's Macquarie Island (54.5 °S, 158.9 °E) during $\sim$ 1 year of the MICRE field campaign. A merged dataset from a 95 GHz (W-band) cloud radar, ceilometers, and atmospheric soundings was constructed to analyze cloud and precipitation property
occurrence frequencies and their dependence on cloud thermodynamics and macrophysics. A 9-year simulation of the NASA GISS ModelE3 ESM was then evaluated against the MICRE observations by extracting outputs at every time step in the grid cell containing Macquarie Island. Forward simulation of GISS-ModelE3 ceilometer and radar variables was performed via the Earth Model Column Collaboratory (EMC$^2$) radar and lidar instrument simulator, accounting for the sensitivities of the instrumentation deployed during MICRE. This approach yielded a comparison of observations and the ESM in a physically
consistent framework. The main conclusions resulting from the observational analysis and the ESM evaluation are as follows:

- Clouds precipitate frequently from liquid cloud base over Macquarie Island ($\sim$ 70% of the time where cloud bases were identified)

- Deeper and colder clouds precipitate more frequently and at a higher intensity than thinner and warmer clouds

- Clouds with colder CTTs precipitate more readily than at warm CTTs even for the same cloud geometric thickness

- Supercooled cloud layers experience more frequent evaporation/sublimation below LCB compared to warm cloud layers

- The GISS-ModelE3 ESM simulation realized a smaller liquid-based cloud occurrence frequency, smaller precipitation occurrence frequency, and greater sub-cloud evaporation/sublimation compared to observations

- GISS-ModelE3 captures observed trends (shape and skewness) in cloud-base precipitation rate distributions whereby precipitation rates increase with decreasing CTT and increasing cloud thickness

- Geometrically and optically thin non-precipitating clouds and fog were similarly common in both observations and GISS-ModelE3

The ESM evaluation demonstrated here followed a framework in which ESM column physics may be evaluated while remaining faithful to the model's physics parameterizations and accounting for instrument sensitivities. Systematic biases observed in GISS-ModelE3 (i.e, lower precipitation occurrence frequencies and more evaporation relative to MICRE observa-
tions) are unlikely to result from thresholding behavior for cloud-base precipitation detection since the biases are consistent



for various minimum radar reflectivity thresholds used to qualify precipitation. Further work is needed in order to better understand these differences as they apply to GISS-ModelE3 and to other ESMs with different physics parameterizations. However, this study demonstrates that long term, ground-based instrumentation can be used as a robust process-level constraint for ESM evaluation when appropriate sensitivities are considered. Such process-driven studies are important to understand cloud and precipitation properties in the present-day atmosphere as well as for perturbed climates and how they may compensate, enhance, or reduce cloud radiative feedbacks in the extratropics.

**Appendix A: Abbreviations and Notation**





**Table A1.** List of abbreviations and notation.

| Abbreviations and Notation | |
| --- | --- |
| $\beta_{\mathrm{att}}$ | 910 nm ceilometer attenuated backscatter; units of m$^{-1}$ sr$^{-1}$ |
| BASTA | Bistatic Radar System for Atmospheric Sciences (Delanoë et al., 2016) |
| CBH | Cloud base height |
| CTH | Cloud top height |
| CBT | Cloud base temperature |
| CTT | Cloud top temperature |
| CEIL | Merged ARM and University of Canterbury ceilometer datasets |
| $D_{\mathrm{min}}$ | Depth below cloud base used for cloud-base precipitation detection; depth above $h_{\mathrm{min}}$ used for surface precipitation detection |
| EMC$^2$ | Earth Model Column Collaboratory instrument simulator (Silber et al., 2022) |
| GISS-ModelE3 | U.S. National Aeronautics and Space Administration (NASA) Goddard Institute for Space Studies ModelE3 |
| $h_{\mathrm{min}}$ | Minimum detectable height of the BASTA radar; set to 150 m AGL |
| LCB | Liquid cloud base |
| $P_{\mathrm{cb}}$ | Cloud-base precipitation occurrence frequency |
| $P_{\mathrm{sfc}}$ | Surface precipitation occurrence frequency; surface is $h_{\mathrm{min}}$ = 150 m AGL |
| $R_{\mathrm{cb}}$ | Cloud-base precipitation rate |
| $R_{\mathrm{sfc}}$ | Surface precipitation rate; surface is $h_{\mathrm{min}}$ = 150 m AGL |
| RH$_{\mathrm{liq}}$ | Relative humidity with respect to liquid water |
| RH$_{\mathrm{sfc}}$ | Relative humidity from surface meteorology station |
| RH$_{\mathrm{ice}}$ | Relative humidity with respect to ice |
| $Z_e$ | W-band (95 GHz) radar reflectivity; units of dBZ |
| $Z_{e,min}$ | Minimum detectable $Z_e$ |
| $\overline{Z_{e,150-250m}}$ | Linearly averaged reflectivity between 150 and 250 m AGL |

## Appendix B: Minimum Detectable BASTA $Z_e$

The BASTA radar reports a $Z_{e,\mathrm{min}}$ of -36 dBZ at 1 km. Fig. B1 shows the theoretical $Z_{e,\mathrm{min}}$ as a black dashed line, which is
calculated assuming irradiance weakens inversely proportional with the square of range, while the light blue line shows the $0.01^{st}$ percentile of BASTA $Z_e$ from the year during MICRE when BASTA was operational. Range gates where the reflectivity as a function of height is less than the theoretical $Z_{e,\mathrm{min}}$ are assumed to be free of hydrometeors. Importantly, we note that the BoM BASTA radar used here as reported by Delanoë et al. (2016) and also used by Mace and Protat (2018a, b) nominally reports a sensitivity of -49 dBZ at 1 km AGL (-36 dBZ at 4 km AGL). However, issues with the low noise amplifier during





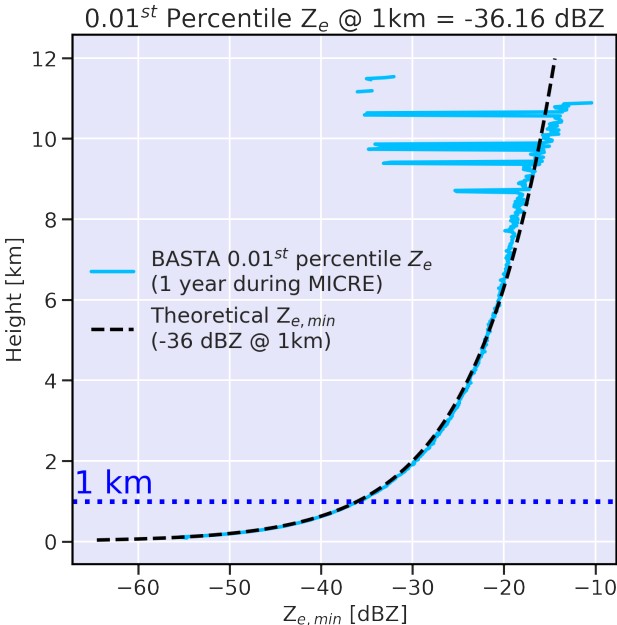

**Figure B1.** Profiles of the theoretical minimum $Z_e$ ($Z_{e,\mathrm{min}}$, black dashed line) for a sensitivity of -36 dBZ at 1 km AGL and the $0.01^{st}$ percentile of BASTA reflectivity (effective $Z_{e,\mathrm{min}}$, solid blue line) as a function of height. BASTA data is composited across the entire year of MICRE.

MICRE degraded the BoM BASTA sensitivity to -36 dBZ at 1 km AGL. We emphasize however that between $h_{\mathrm{min}}$ (150 m) and 1 km AGL, BASTA detects $Z_e$ values down to -55 dBZ.

**Appendix C: Addressing Potential Biases in LCB Height Detections**

Following from the finding in Silber et al. (2018) that the ARM ceilometer tends to underestimate true LCB height by 36-50 m relative to other observing methods, $P_{\mathrm{cb}}$ is recalculated by offsetting the native CEIL-identified CBH downwards by 25 or 50

m (1 to 2 BASTA range gates). $P_{\mathrm{cb}}$ is shown for these modified calculations in Fig. C1, where lowering the CBH by 25 (50) m decreases the total $P_{\mathrm{cb}}$ at the highest sensitivity by 2 (5)%. Sensitivities to $Z_{e,\mathrm{min}}$ and $D_{\mathrm{min}}$ remain consistent with these offset CBHs. In general, offsetting the cloud base decreases the total $P_{\mathrm{cb}}$, but the sensitivity is small.

**Appendix D: Sounding RH and Ceilometer CBH Comparison**

Evaluation of ceilometer CBHs was performed by co-locating in time with soundings released at nominally 12 hour intervals

during MICRE. Fig. D1 shows a joint histogram of $\mathrm{RH}_{\mathrm{liq}}$ and temperature at heights where the ceilometer recognized a CBH within 20 minutes after a sounding release time. There is a clear maximum in frequency for $\mathrm{RH}_{\mathrm{liq}} > 98\%$. Following from





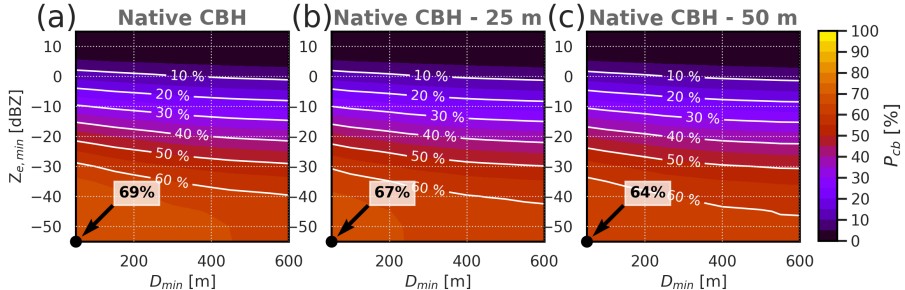

**Figure C1.** Cloud-base precipitation occurrence frequency ($P_{\text{cb}}$, contours and color fill) as a function of $Z_{e,\text{min}}$ threshold (ordinate) and vertical resolution ($D_{\text{min}}$, abscissa) for all cloud layers using (a) the native CEIL-recognized CBH, (b) the native CBH offset by 25 m, and (c) the native CBH offset by 50 m. The black circles in the bottom left-hand corner of each panel represents the BASTA $Z_{e,\text{min}}$ and $D_{\text{min}} = $ 50 m (2 range gates).

Silber et al. (2021) and assuming an $RH_{\text{liq}}$ uncertainty of 5%, we consider a liquid-bearing cloud layer to have $RH_{\text{liq}} > 95\%$ in the sounding. In Fig. 1, altitude ranges where $RH_{\text{liq}} > 95\%$ are identified by transparent purple shading in the sounding profile and in the BASTA time-height series, with the sounding-based CBH and CTH shown as dark purple lines. Fig. 1d
shows that this $RH_{\text{liq}}$ threshold appropriately identifies a sounding-based CBH where the ceilometer identifies CBH, and that the sounding-identified CTH is correctly located at the top of the radar reflectivity hydrometeor-containing layer. The low-frequency scatter of ceilometer CBHs with $RH_{\text{liq}} < 95\%$ in Fig. D1 is due to heterogeneity in the vertical placement of the liquid layer that causes spatiotemporal discrepancies between the cloud environment sampled by CEIL and by the sounding. Overall, 66 (80)% of soundings with colocated CEIL-identified CBHs obtained $RH_{\text{liq}}$ values $> 95$ (90)%.

**Appendix E: Supercooled Partitioning Dependence on $D_{\text{min}}$**

The partitioning of supercooled versus warm-based (i.e., warm + partially supercooled) cloud layers is a strong function of $D_{\text{min}}$ (Fig. E1). At $D_{\text{min}} = 50$ m, $\sim 55\%$ of detected clouds are supercooled while $\sim 45\%$ of clouds have warm CBTs. At $D_{\text{min}} = 600$ m, the fraction of cloud layers identified as supercooled increases (decreases) to $\sim 85\%$ (15%) for supercooled (warm-based) clouds. This is due to the higher $D_{\text{min}}$ threshold limiting the number of clouds that can be detected below the
minimum detectable CBH (i.e., $D_{\text{min}} + h_{\text{min}}$). For a minimum CBH of 750 m, a large fraction of warm-based cloud layers are omitted from the analysis and the total sample size of clouds capable of cloud-base precipitation detection decreases.

**Appendix F: Fog Case Studies**

Generalized cloud morphologies are recognized during MICRE as representative of fog, where two primary cloud environments are demonstrated in Figs. F1 and F2. The first case (Fig. F1) is representative of a moderate-to-heavy precipitation event with
intermittent periods of precipitation breaks. In these intermittent periods (e.g., $\sim 0600$ UTC), a shallow cloud layer is notable



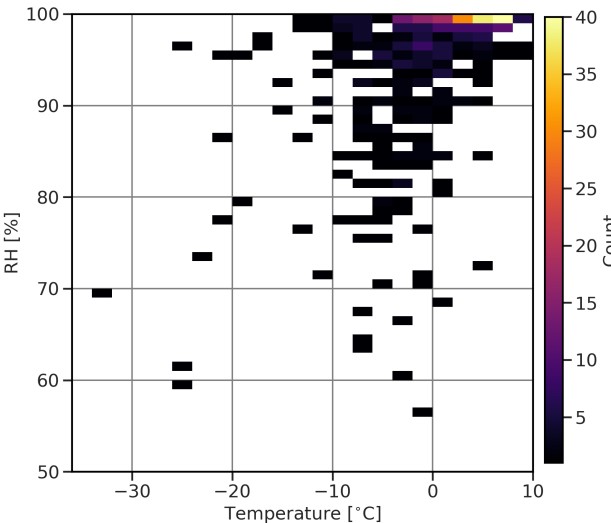

**Figure D1.** Joint histogram of temperature and relative humidity (RH) from soundings at the ceilometer-recognized CBH for all valid soundings during MICRE.

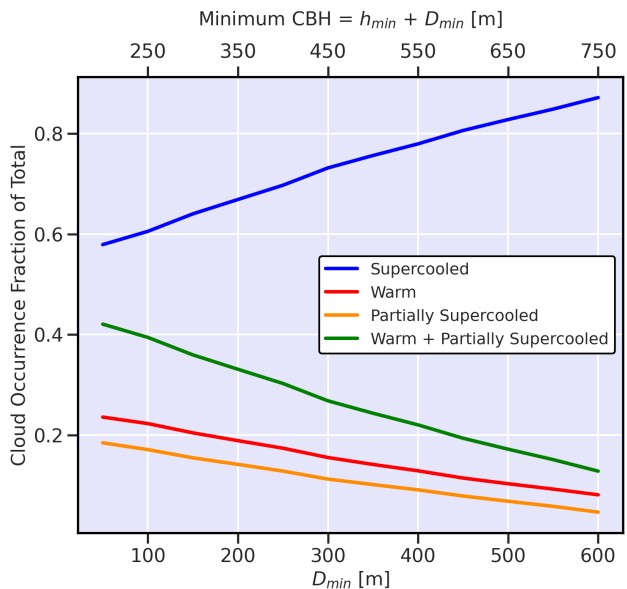

**Figure E1.** Fraction of total detected clouds able to be identified as precipitating, distributed among supercooling (colors), as a function of vertical resolution ($D_{min}$). Top axis is the minimum CBH, which is equivalent to $h_{min}$ (=150 m) + $D_{min}$.





in the radar reflectivity that reaches 400 m AGL (Fig. F1c,d). The $RH_{sfc}$ exceeds 95% during this time period (Fig. F11f) and the two soundings released during the event show completely saturated layers through at least 1 km AGL (Fig. F1a,b). The University of Canterbury ceilometer $\beta_{att}$ (Fig. F1e) shows a persistent period from 0000 to 1400 UTC with CBHs very close to the surface (within the BASTA "blind zone") where the signal is entirely attenuated above $\sim 125$ m AGL.

The second example is a more traditional fog layer (Fig. F2). The CEIL $\beta_{att}$ profiles begin with enhanced values $> 10^{-6}$ m$^{-1}$ sr$^{-1}$ without coincident radar reflectivity, which may be deliquescenced aerosol (haze), before developing a surface-based cloud where $\beta_{att}$ values exceed $10^{-4}$ m$^{-1}$ sr$^{-1}$ and radar echoes develop ($\sim 1900$ UTC on 22 May). $RH_{sfc}$ then exceeds 95% and the last sounding shows a liquid-saturated layer up through 800 m AGL. The CBHs in this case are not as consistent as in Fig. F1, with CBHs often being detected up to 400 m AGL. Note also instances (e.g., $\sim 0900$ UTC on 23 May) where shallow
convection appears to rise out of the fog layer.

Regardless of the formation mechanism, these $\beta_{att}$ profiles and their physical implications account for a large portion of cloud bases identified by CEIL (14%, see Fig. 9). Although such profiles may be regarded as contamination of the ceilometer signal, they are coincident with $RH_{sfc} > 95\%$, suggesting a prevalence of fog over this SO site with true cloud bases near the surface, and thus the relevant physical formation mechanisms should be represented by model physics.

**Appendix G:  GISS-ModelE3 Sensitivities of Cloud and Precipitation Occurrence Frequency to Thresholding Behavior**

Cloud and precipitation occurrence frequencies may be sensitive to certain thresholding behavior in the model. LCB detection in GISS-ModelE3 is performed by identifying the lowest grid cell in altitude where CLWC exists. An arbitrary lower threshold for the statistics discussed here is found to be unnecessary for representing cloud occurrence frequency, which only decreases by a few percent between grid-cell mean CLWC values of $10^{-9}$ g m$^{-3}$ to $10^{-4}$ g m$^{-3}$ (Fig. G1a). Similarly, the cloud
occurrence frequency is shown to be insensitive for $\beta_{att} < \sim 10^{-5}$ m$^{-1}$ sr$^{-1}$ (top axis of Fig. G1a).

The detection of precipitation relies on the existence of a precipitating hydrometeor species within the grid cell identified as cloud base, no matter how small the $R_{cb}$ is in that grid cell. However, Fig. G1b shows that the precipitating fraction only decreases by $\sim 2.5\%$ for a range of $R_{cb}$ from $10^{-12}$ mm hr$^{-1}$ to $10^{-6}$ mm hr$^{-1}$. This implies that the precipitation occurrence frequency is also not very sensitive to relevant minimum $R_{cb}$ thresholds we expect to observe in nature.

*Data availability.*  Department of Energy (DOE) Atmospheric Radiation Measurement (ARM) program ceilometer data (doi:10.5439/1181954) and Australian BoM surface meteorology station data (doi:10.5439/1597382) are available through the DOE ARM data archive (https://adc.arm.gov/). BASTA radar data (doi:10.26179/5d91836ca8fc3) and the University of Canterbury ceilometer data (doi:10.26179/5d91835e2ccc3) are accessible through the Australian Antarctic Division's Data Centre (https://data.aad.gov.au/metadata/records/AAS_4292_Macquarie_Cloud_Radar and https://data.aad.gov.au/metadata/records/AAS_4292_Macquarie_Ceilometer, respectively). Upper air soundings from the Australian
BoM are available via online request at https://data.aad.gov.au/metadata/records/Antarctic_Meteorology. VISST-derived pixel-level products from the Himawari-8 satellite are available on the ARM Data Discovery website (https://adc.arm.gov/discovery/#/results/site_code::mcq). The Earth Model Column Collaboratory (EMC$^2$) software package is available at https://github.com/columncolab/EMC2. The merged

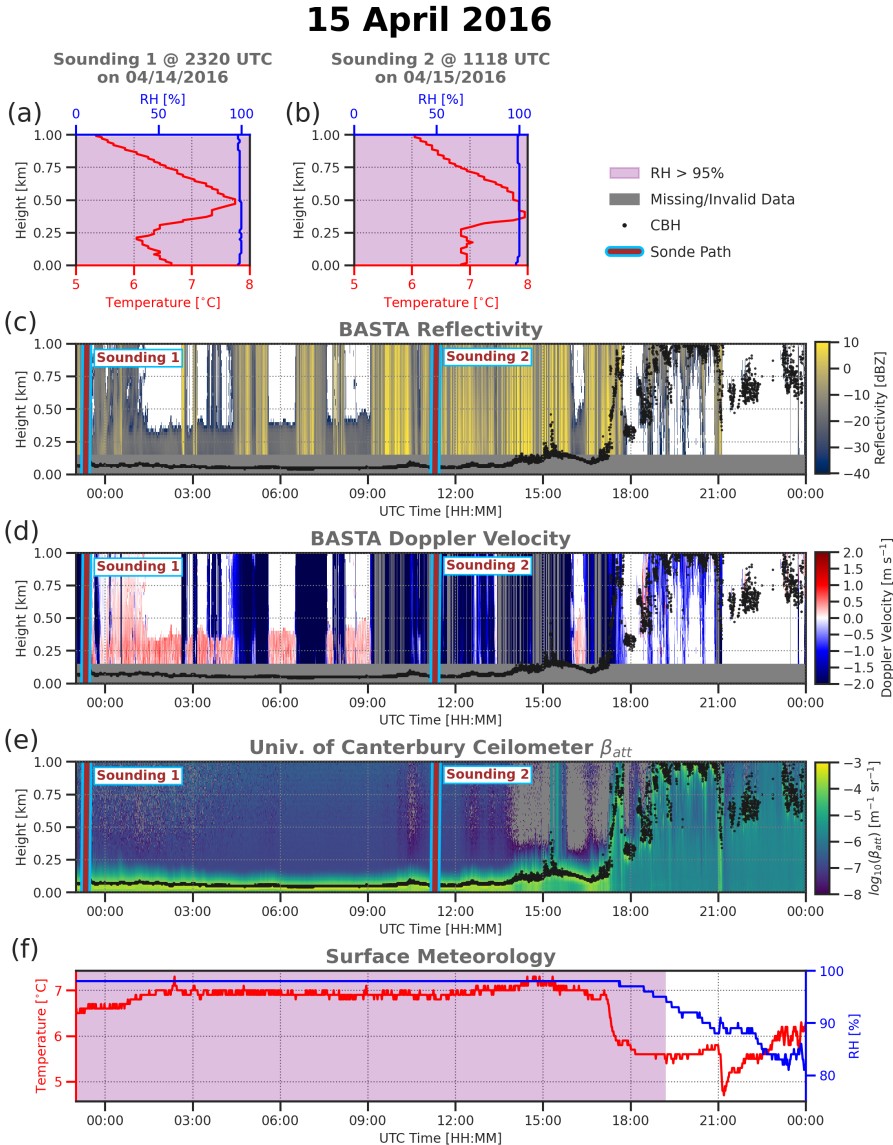

**Figure F1.** Summary of fog event that occurred during moderate to heavy precipitation on 15 April 2016 showing (a) temperature and RH for a sounding released at 2320 UTC on 14 April 2016, (b) as in (a) but for a sounding released at 1118 UTC on 15 April 2016, (c) 24-hr time-height series of BASTA radar reflectivity (d) as in (c) but for BASTA Doppler velocity, (e) as in (c) but for $\beta_{att}$ from the University of Canterbury ceilometer, and (f) a time series of $RH_{sfc}$ and $T_{sfc}$ from the surface meteorological station. In (a), (b), and (f), times/heights where RH > 95% are shaded in purple. CBHs from CEIL are given as black dots in panels (c)-(e). Sounding release times are marked in (c)-(e) via brown lines with a light blue outline.



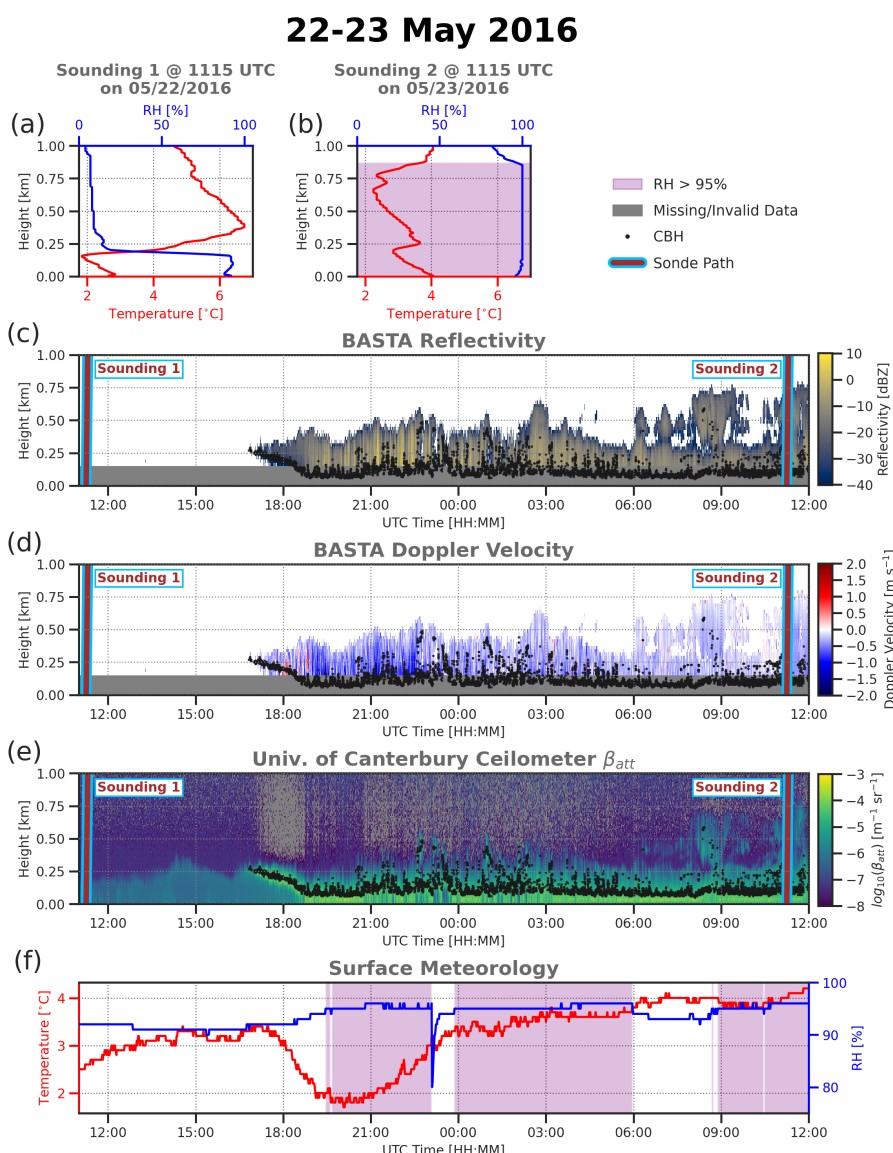

**Figure F2.** As in Fig. F1 but for a fog event from 22-23 May 2016 where the first sounding was released at 1115 UTC on 22 May and the second sounding was released at 1115 UTC on 23 May.





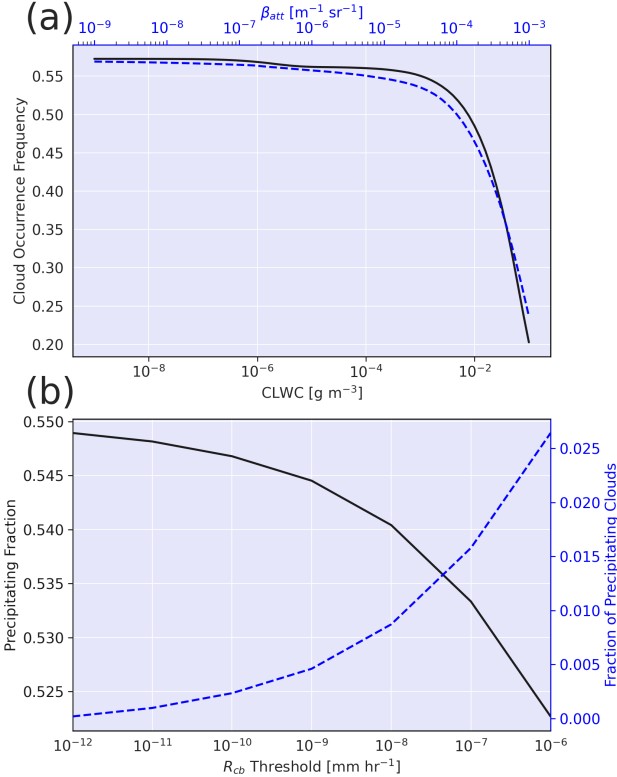

**Figure G1.** Sensitivities of (a) cloud occurrence frequency to thresholding of grid-cell mean cloud liquid water content (CLWC, black solid line) and ceilometer attenuated backscatter ($\beta_{att}$, blue dashed line) and (b) precipitating fraction to minimum $R_{cb}$ threshold (black line). In (b), the fraction of precipitating clouds as a function of $R_{cb}$ threshold is shown as a blue dashed line.

MICRE dataset will also be made available upon manuscript publication as a U.S. DOE ARM Principal Investigator (PI) data product (https://adc.arm.gov/discovery/). Code used for processing and scripts used to make all figures are available at https://github.com/NASA-
GISS/micre_stanford-acp2023.

*Author contributions.* MS, IS, and AF conceptualized the study. MS prepared and analyzed the observational datasets with assistance from IS. IS performed the GISS-ModelE3 simulations with assistance from AA and supported use of the EMC$^2$ software. MS performed the model evaluation and drafted the manuscript, with subsequent input from all co-authors. AA contributed to interpretation of results. GC and JM aided in interpreting results as they relate to GISS-ModelE3 evaluation and cloud-climate feedbacks. AP provided the BASTA radar data.
SA and AM provided the University of Canterbury ceilometer data. SA conceptualised the overall science goals for MICRE, secured funding and logistical support, and led the field deployment to Macquarie Island.



*Competing interests.* The authors declare that they have no competing interests.

*Financial support.* This work was supported by the Office of Science (BER), U.S. Department of Energy (DOE), under Agreements DE-SC0016237 and 89243021SSC000078, and the NASA Modeling, Analysis and Prediction Program. I.S. was supported by DOE grant DE-SC0021004.

*Acknowledgements.* We thank Emily Tansey and Roger Marchand for their valuable discussions. Computing resources supporting this work were provided by the NASA Center for Climate Simulation (NCCS) at Goddard Space Flight Center. Technical and logistical support for MICRE was provided by the Australian Antarctic Division through Australian Antarctic Science Projects 4292 and 4387, and we thank Andrew Klekociuk, John French, Peter de Vries, Terry Egan, Nick Cartwright, and Ken Barrett for all of their assistance.

*Review statement.*



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
