# Peer review of "Earth System Model Evaluation of Cloud and Precipitation Occurrence for Supercooled and Warm Clouds over the Southern Ocean's Macquarie Island"

_EGUsphere, 2023_

## Referee Comment (RC1)

Review of manuscript egusphere-2023-170

**Title:** Observed Process-level Constraints of Cloud and Precipitation Properties over the Southern Ocean for Earth System Model Evaluation

**Authors:** McKenna W. Stanford, Ann M. Fridlind, Israel Silber, Andrew S. Ackerman, Greg Cesana, Johannes Mülmenst.dt, Alain Protat, Simon Alexander, and Adrian McDonald

Summary:

In this manuscript, Stanford and coauthors examine the occurrence of cloud and (below-cloud) precipitation over the SO using cloud radar, two lidar ceilometers, and balloon-borne soundings measurements collected during MICRE. They compare the measurements and derived occurrences (for various low cloud types) with simulations from the GISS-Model3E using, radar and lidar instrument simulators.

Recommendation: Publish in with minor revisions.

I very much enjoyed reading this article. The manuscript is well written and well organized, and the data analysis has been done with care. Frankly, I think the manuscript is perfectly publishable in its present form. What comments I have, are frankly being rather picky.

This is Roger Marchand. I normally do not identify myself in the review processes, but I do so here for a couple of reasons:

First, I only realized when I got to the end of the manuscript that I am acknowledge for some past conversations I have had with the authors. Had I realized this, I probably would have declined to be a reviewer. I don't think my review is biased, but I think it is appropriate to note this fact. Regardless, thank you for the acknowledgment.

Second, as you will see below, I have some specific suggestions as regards ongoing work by my students.

Comments:

1) Title: Constraint on Physical Processes, "Properties", Representativeness of the site

I do not care for the title.

(i) While I agree that the observations have the potential to provide constraints on physical processes, this is not actually demonstrated or explored in this manuscript. I think doing such would make for a nice paper, and I hope that you will do such as part of future work. But it seems like overreach to simply claim the observables constrain processes, and problematic to provide no information on what processes or to what

degree they might be constrained.

(ii) "Properties" is rather vague. The manuscript is focused on cloud and precipitation occurrence for warm, supercooled and partially supercooled clouds. Why not put these terms in the title? I think this will make the paper easier to find for people interest in how supercooled and warm clouds differ.

(iii) This bothers me the least, but with a high degree of confidence there is a significant latitudinal dependence in some of the occurrence statistics which are (or maybe) comparable to the differences you note between the observations and model output.

Might I suggest a title change to something such as:

"Earth System Model Evaluation using Cloud and Precipitation Occurrence for Supercooled and Warm Clouds over the Southern Ocean (based on observations during MICRE)."

Or

"A Comparison of GISS-Model3E and MICRE Observed Cloud and Precipitation Occurrence for Supercooled and Warm Clouds over the Southern Ocean using Instrument Simulators."

2) Depolarization lidar phase retrievals (L. 165)

You note in the manuscript that processing of the depolarization lidar has been a problem. This is true. But my student, Emily Tansey has been working on this for the last year and I think we have the problems "worked out" as best we can. We are submitting a publication on this in two weeks. My point here is not that you should delay publication of this manuscript, only that I would prefer if you wrote something to the effect of "The depolarization lidar data have calibration stability and other problems which are being corrected and will be released soon [manuscript in preparation, Tansey and Marchand, University of Washington]

3) The Comstock, Z-R Relationship (Line 233)

You may be interested to know that I have another student (Litai Kang) looking at Z-R relationships based on SOCRATES aircraft data. In spite of the fact that the Comstock relationship was based on VOCALS data (subtropical StCu), it seems to hold surprisingly well for the SO. I list our equations below. We are working to get this published, but I suggest you add something to the effect of "An examination of in situ aircraft data from SOCRATES finds the Comstock relationship holds well for drizzle falling from SO stratocumulus [personal communication or manuscript in preparation, Kang and Marchand, University of Washington]."

Based on in situ data (~ 150m above the surface) ➜ $Z = (63.8 \pm 47.1) \, R^{(1.3 \pm 0.05)}$

Based on SOCRATES radar-lidar retrievals ➔ $Z = (31.6 \pm 1.4) R^{1.4}$

(The R^1.4 is an interesting story for another day.  And I think the "a" coefficient appears to be a bit higher because the number of drizzle droplets appears to be higher on average for the SOCRATES measurements as compared to VOCALS, and I suspect this is climatologically true but pretty small datasets, really.).

4)  Seeder feeder?

Having worked with these data extensively, I know there is not a tremendous amount of seeder feeder generated precipitation coming from (through) low clouds.  Nonetheless, I wonder if you have done anything to filter such out?   Have you looked at your statistics when limiting to single-layer clouds (if yes, does it matter)?  I am bit concerned about this because you state on Line 140, that you looked at only the lowest cloud (and are not leaving much room for ambiguity in the LCB – see also comment line 356).

Specific comments:

Line 27.   What about CMIP6?   The material here seems a bit dated and there are several papers out now looking at radiative biases in CMIP6.   I am somewhat amused by the fact that there are 99 or so papers by coauthor Cesana cited here, but not his recent paper looking at the SO radiative bias in CMIP6 models.   You might also consider Lauer et al. 2023.

Line 140.  As you note a few lines below, CBH here means the peak in the backscatter. The algorithm finds the peak well, but I think one might debate whether the peak is the best way to define cloud base.  In fact, you discuss this in more detail later in the manuscript (or reference Silber's 2018 paper were he does).  Perhaps move material form line 147 up and note the peak is generally found with a small RMS uncertainty (+/- 5 m) for liquid clouds?

Line 154.  Merged in what way?  Via a nearest-neighbor interpolation?  Averaging the ceilometer in some time window?  I did the later in my work with these data, in part because it provides an easy way to combine the ARM and UC data.

Line 219.  Using a threshold (-55 dBZ) that is below the radar minimum detectable signal is not a problem per se, but you should not expect the results to be the same as what you would get from a radar that had a minimum detectable signal of -55 dBZ over the entire below-cloud region.  That is there would be more precipitation at the lower thresholds with a more sensitive radar.  I don't think this would qualitatively change your points, but I think you could be clearer on this.   Perhaps in the appendix (see also Line 671).

Line 324.  "… Rcb was also found to increase for decreasing CTT while controlling for cloud thickness (not shown) …".   Why not add a panel to show this?   I think this is a valuable point.

*Line 356.  I think this should be 7% to 27%.  If 27% have a LCB without a radar detection (at the location of cloud base) but 20% DO have a radar detection within 100 m, might it be the case that the radar is simply not able to see the bottom portion of the cloud and therefore only 7% of the time there is a "lidar only" detected cloud?   (In my own work, I assume LCBs within 100m of radar detection are the same cloud and thus report a value closer to 10% for lidar-only detected clouds).

Line 394.  The Comstock relationship was not developed using fog, which I believe is known to be composed of droplets that are smaller on average than that found for most clouds.   I am not sure this is well justified.

Line 503. Too much stratocumulus?  I would say this is debatable, since the study in question is based on ISCCP obs and ISCCP-clustering.  (1) While there is too much of the "stratocumulus" cluster, I am not sure there is too much actual stratocumulus cloud (in total) and (2) one needs to be careful with ISCCP measures for low cloud occurrence (such is lower in ISCCP as compared with say MISR or CALIPSO/CloudSat).   Perhaps change to "… and MAY now simulate …".

Line 505.  "… 57% in GISS-ModelE3 compared to 76% in MICRE".  So is it fair to conclude that cloud occurrence (and presumably stratocumulus) remains a bit low in GISS-ModelE3 (in spite of the previous statement)?   Perhaps add a direct comment rather than leave such for the reader to piece together.

Line 531. "if satellite observations underestimate precipitation occurrence frequency relative to collocated ground-based observations … " Don't the next two sentences demonstrate that this "middle point" IS true.    Perhaps you should break this sentence apart and comment on the degree to which each of these might be true?   As-is, you give three reasons, and say nothing about two of them, leaving the reader to infer what they will.

Line 671.  "...and 1 km"?   I'm confused by this comment.  Only near h_min is the sensitivity this good.   Near 1 km you are near -36 dBZ, so a radar volume with -45 dBZ worth of precipitation at 1 km wont be detected.

Line 673.  "underestimate", meaning the vendor LCB is too low in altitude (units are given in m not Pa)?   I think you mean the CEIL LCB tends to be high !?!  Perhaps rephrase to make clear.

---

## Author Comment (AC1)

**We thank the two reviewers for their thorough comments and suggestions and feel that they have helped to greatly improve the manuscript. Below, we provide detailed responses to each individual comment. Any added or modified text in the manuscript as a result of the comment response is provided as line numbers of the revised manuscript and quotation of the added/modified material. Importantly, we note that following the first comment from Reviewer 1, we have decided to change the title to "Earth System Model Evaluation of Cloud and Precipitation Occurrence for Supercooled and Warm Clouds over the Southern Ocean's Macquarie Island."**

**Reviewer 1**

Summary: In this manuscript, Stanford and coauthors examine the occurrence of cloud and (belowcloud) precipitation over the SO using cloud radar, two lidar ceilometers, and balloonborne soundings measurements collected during MICRE. They compare the measurements and derived occurrences (for various low cloud types) with simulations from the GISS-Model3E using, radar and lidar instrument simulators. Recommendation: Publish in with minor revisions. I very much enjoyed reading this article. The manuscript is well written and well organized, and the data analysis has been done with care. Frankly, I think the manuscript is perfectly publishable in its present form. What comments I have, are frankly being rather picky. This is Roger Marchand. I normally do not identify myself in the review processes, but I do so here for a couple of reasons: First, I only realized when I got to the end of the manuscript that I am acknowledge for some past conversations I have had with the authors. Had I realized this, I probably would have declined to be a reviewer. I don't think my review is biased, but I think it is appropriate to note this fact. Regardless, thank you for the acknowledgment. Second, as you will see below, I have some specific suggestions as regards ongoing work by my students.

We thank Dr. Roger Marchand for his careful review and detailed comments. Dr. Marchand's clear understanding of the MICRE campaign and associated instrumentation have helped us to adjust specific reasonsings and support our conclusions from this study, leading to an improved manuscript that expands on the techniques we carried out.

Comments:

1) Title: Constraint on Physical Processes, "Properties", Representativeness of the site
I do not care for the title.
(i) While I agree that the observations have the potential to provide constraints on physical processes, this is not actually demonstrated or explored in this manuscript. I think doing such would make for a nice paper, and I hope that you will do such as part of future work. But it seems like overreach to simply claim the observables constrain processes, and problematic to provide no information on what processes or to what degree they might be constrained.
(ii) "Properties" is rather vague. The manuscript is focused on cloud and precipitation

occurrence for warm, supercooled and partially supercooled clouds. Why not put these terms in the title? I think this will make the paper easier to find for people interest in how supercooled and warm clouds differ.

(iii) This bothers me the least, but with a high degree of confidence there is a significant latitudinal dependence in some of the occurrence statistics which are (or maybe) comparable to the differences you note between the observations and model output. Might I suggest a title change to something such as:

"Earth System Model Evaluation using Cloud and Precipitation Occurrence for Supercooled and Warm Clouds over the Southern Ocean (based on observations during MICRE)."

Or

"A Comparison of GISS-Model3E and MICRE Observed Cloud and Precipitation Occurrence for Supercooled and Warm Clouds over the Southern Ocean using Instrument Simulators."

**Response:** These are good points. In particular, we previously considered point #3 as we recognize the likelihood of a latitudinal dependence. Regarding point #1, we consider precipitation occurrence frequency to be a simple constraint on the precipitation process–a robust binary designation of whether or not a precipitation process is active (e.g., see Mülmenstädt et al., 2021), with some insight given by the supercooling partitioning and the potential for mixed phase processes to modulate that occurrence frequency (see also response to Reviewer 2's Comment #5). Since GISS-ModelE3 and many other ESMs partition precipitating and cloud hydrometeor species and consider their own set of processes based on that classification, we do think that having a range of precipitation occurrence frequencies operates as a process-level constraint for models. However, we also recognize that we are not explicitly distinguishing between any specific precipitation processes. Nonetheless, we do think there is some important information left out of the current title, particularly the lack of explicitly mentioning an ESM evaluation. We have therefore revised the title to the following: "Earth System Model Evaluation of Cloud and Precipitation Occurrence for Supercooled and Warm Clouds over the Southern Ocean's Macquarie Island"

2) Depolarization lidar phase retrievals (L. 165)

You note in the manuscript that processing of the depolarization lidar has been a problem. This is true. But my student, Emily Tansey has been working on this for the last year and I think we have the problems "worked out" as best we can. We are submitting a publication on this in two weeks. My point here is not that you should delay publication of this manuscript, only that I would prefer if you wrote something to the effect of "The depolarization lidar data have calibration stability and other problems which are being corrected and will be released soon [manuscript in preparation, Tansey and Marchand, University of Washington]

**Response:** Thanks so much for pointing this out. The point of the statement was to justify our intentional neglect of the dataset and acknowledging that the lidar was there. We have modified this sentence to now read the following on lines 177-179 of the revised manuscript: "We also

note that although a polarization lidar was present during the MICRE campaign, the data have calibration stability and other problems that prevented its use in this study, but are being corrected and will be released soon (Tansey et al., submitted)."

3) The Comstock, Z-R Relationship (Line 233)
You may be interested to know that I have another student (Litai Kang) looking at Z-R relationships based on SOCRATES aircraft data. In spite of the fact that the Comstock relationship was based on VOCALS data (subtropical StCu), it seems to hold surprisingly well for the SO. I list our equations below. We are working to get this published, but I suggest you add something to the effect of "An examination of in situ aircraft data from SOCRATES finds the Comstock relationship holds well for drizzle falling from SO stratocumulus [personal communication or manuscript in preparation, Kang and Marchand, University of Washington]."
Based on in situ data (~ 150m above the surface) $\rightarrow$ Z = (63.8 ± 47.1) R $^{(1.3 \pm 0.05)}$
Based on SOCRATES radar-lidar retrievals $\rightarrow$ Z = (31.6 ± 1.4) R $^{(1.4)}$
(The R^1.4 is an interesting story for another day. And I think the "a" coefficient appears to be a bit higher because the number of drizzle droplets appears to be higher on average for the SOCRATES measurements as compared to VOCALS, and I suspect this is climatologically true but pretty small datasets, really.).

**Response:** Thanks for directing this information to us. It's great to hear that this work is being pursued. We have added a sentence on lines 241-243 of the revised manuscript to state this: "An examination of in situ aircraft data from the Southern Ocean Clouds, Radiation, Aerosol Transport Experimental Study (SOCRATES; McFarquhar et al., 2021) finds the Comstock et al. (2004) relationship holds well for drizzle falling from SO stratocumulus (manuscript in preparation, Kang and Marchand, University of Washington)."

4) Seeder feeder?
Having worked with these data extensively, I know there is not a tremendous amount of seeder feeder generated precipitation coming from (through) low clouds. Nonetheless, I wonder if you have done anything to filter such out? Have you looked at your statistics when limiting to single-layer clouds (if yes, does it matter)? I am bit concerned about this because you state on Line 140, that you looked at only the lowest cloud (and are not leaving much room for ambiguity in the LCB – see also comment line 356).

**Response:** We find that only ~7.5 % of ceilometer-recognized CBH profiles contained more than one CBH detection and thus this does not significantly impact results. We have added the following language on lines 284-286 of the revised manuscript to describe this sensitivity: "We note that limiting profiles to those containing only one CEIL-recognized CBH (single layer clouds) changed $P_{cb}$ by < 1 %, therefore likely mitigating significant influence of seeder-feeder mechanisms (e.g., He et al., 2022) to the extent that the ceilometer is not fully attenuated beyond the lowest cloud layer."

**Specific comments:**

Line 27. What about CMIP6? The material here seems a bit dated and there are several papers out now looking at radiative biases in CMIP6. I am somewhat amused by the fact that there are 99 or so papers by coauthor Cesana cited here, but not his recent paper looking at the SO radiative bias in CMIP6 models. You might also consider Lauer et al. 2023.

**Response:** Thank you for bringing this to our attention. Indeed, differences between CMIP5 and CMIP6 generations are notable, and we have added a short discussion of this on lines 29-35 of the revised manuscript to state the following: "CMIP6 models improved this bias to some degree (e.g., Schuddeboom et al., 2021; Cesana et al., 2022), but low- and mid-level clouds at latitudes higher than 55 ˚S were found to still produce a low bias in reflected SW radiation compared to satellite observations (e.g., Mallet et al., 2023), likely due to poor phase representation in the dominant supercooled liquid cloud regime (Cesana et al., 2022). Furthermore, the equilibrium climate sensitivity (ECS) has increased from CMIP5 to CMIP6 generations, primarily due to stronger positive low cloud feedbacks (Zelinka et al., 2020) that may contribute to increased high-biased sea surface temperatures in CMIP6 compared to CMIP5 (e.g., Zhang et al., 2023)."

Line 140. As you note a few lines below, CBH here means the peak in the backscatter. The algorithm finds the peak well, but I think one might debate whether the peak is the best way to define cloud base. In fact, you discuss this in more detail later in the manuscript (or reference Silber's 2018 paper were he does). Perhaps move material form line 147 up and note the peak is generally found with a small RMS uncertainty (+/- 5 m) for liquid clouds?

**Response:** We recognize that the peak in backscatter is not a perfect definition of cloud base. This motivated the analysis performed in Appendix D where ceilometer-identified CBHs were co-located in time with available soundings, showing that the majority of ceilometer-identified cloud base heights coincided with sounding RH > 95%, which is considered a threshold for sounding-identified cloud layers given a 5% uncertainty in sounding RH (Silber et al. 2021). Furthermore, Appendix C addressed the potential for high-biased CBHs (assuming the true cloud base is 25-50 m above the peak backscatter CBH), and we found it did not significantly impact results regarding precipitation occurrence frequency.

Defining liquid cloud base was unexpectedly (to us) one of the most significant challenges in this study. While some tests were performed to use the attenuated backscatter from the ceilometers to define CBH differently, it was ultimately determined that retaining the original CBH product led to the most consistent interpretation. Moreover, attempts were made to calibrate the ceilometer attenuated backscatter using the autocalibration method of O'Connor et al. (2004). This was important for fog identification since the fog algorithm includes a threshold of the attenuated backscatter coefficient. However, we found few cases where calibration was appropriate and instead decided to quantify fog by accounting for uncertainty in the ceilometer attenuated backscatter calibration. The following additional discussions were thus added to the manuscript:

Lines 154-158: " We note that attenuated backscatter was not calibrated in this study since CBH is provided by instrument firmware. Uncalibrated or "apparent" $\beta_{att}$ is shown in Fig. 1f only for demonstration of peak $\beta_{att}$ associated with cloud base. However, attenuated backscatter is used to evaluate near-surface clouds in Section 3.4.2, where sensitivities to instrument calibration are considered and discussed."

Lines 404-416: "There are several caveats to this detection method. First, only profiles with a valid CBH detection below 250 m AGL are considered, therefore neglecting any profiles where fog may be detectable using $\beta_{att}$ alone. Second, $\beta_{att}$ is uncalibrated. To explore the sensitivity to this, calibration factors were applied to all near-surface CBH profiles (e.g., O'Connor et al., 2004, Hopkin et al., 2019; Kuma et al., 2021). Calibration factors were guided by literature (Kuma et al., 2021) and by applying the lidar autocalibration method described by O'Connor et al. (2004) for optically thick non-precipitating stratocumulus, though we note that few cases were found to be appropriate for calibration with this method in this dataset. For a cloud-base $\beta_{att}$ threshold of $10^{-4.5}$ m$^{-1}$ sr$^{-1}$ used for fog identification, calibration factors ranging from 1-4 yielded fog occurrence frequencies relative to all near-surface clouds that ranged from 69-82 %. Sensitivity to calibration factors increased with increasing cloud-base $\beta_{att}$ thresholds, and the fog occurrence frequency in general was more sensitive to this threshold than to calibration. Given these multiple uncertainties, we do not formally attempt to calibrate $\beta_{att}$ in this study, but note that future work concerning surface-based fog detection over the Southern Ocean should consider all profiles with valid $\beta_{att}$ (regardless of valid CBH detection) and should pursue calibration methods appropriate for fog."

Regarding the discussion of peak backscatter representing LCB, we moved material from former line 147 closer to material from former line 140, now on lines 147-149 in the revised manuscript. We already noted that the ceilometer had a CBH uncertainty of +/- 5 m, which we've tried to make more clear that this is associated with an uncertainty in peak backscatter.

Line 154. Merged in what way? Via a nearest-neighbor interpolation? Averaging the ceilometer in some time window? I did the later in my work with these data, in part because it provides an easy way to combine the ARM and UC data.

**Response:** We used a nearest neighbor approach in both time and space. The nearest time requires the difference between a ceilometer CBH retrieval and a BASTA time step to not exceed 12 seconds. The altitude of the CBH retrieval then lies within or on the edge of a BASTA range gate. If the CBH lies on the edge of a range gate, both the above and below range gates are evaluated for coincident radar reflectivity. Since the workflow targeted analysis on the native BASTA time grid of 12 seconds, we did not want to average. We have added the following brief description on lines 165-167 of the revised manuscript: "Cloud base heights are interpolated with a nearest neighbor approach in time and space, where the nearest time cannot exceed 12 seconds from a BASTA time stamp and the nearest heights lie within or on the edge of a valid BASTA range gate."

Line 219. Using a threshold (-55 dBZ) that is below the radar minimum detectable signal

is not a problem per se, but you should not expect the results to be the same as what you would get from a radar that had a minimum detectable signal of -55 dBZ over the entire below-cloud region. That is there would be more precipitation at the lower thresholds with a more sensitive radar. I don't think this would qualitatively change your points, but I think you could be clearer on this. Perhaps in the appendix (see also Line 671).

**Response:** This is a good point that we agree should be made clear, and Appendix B is a convenient place to do so. We have added the following discussion on lines 721-724 in Appendix B of the revised manuscript: "We emphasize that while BASTA detects $Z_e$ values down to -55 dBZ near $h_{min}$ (150 m AGL), the sensitivity below 1 km AGL decreases rapidly with increasing range. Therefore, precipitation detection throughout the lowest 1 km AGL is not the same as a more sensitive radar with a minimum detectable signal of ~ -55 dBZ over the entire 1-km depth."

Regarding Fig. 2, this reasoning is why there is little to no differences in precipitation occurrence frequencies between -55 dBZ and ~ -36 dBZ besides for warm-only clouds (see also Fig. 10), which we think is important since most of these warm cloud bases lie below 1 km AGL.

Line 324. "… Rcb was also found to increase for decreasing CTT while controlling for cloud thickness (not shown) …". Why not add a panel to show this? I think this is a valuable point.

**Response:** We did originally provide this as a separate figure, but felt the manuscript was already quite dense, so we decided to exclude this information to maintain focus.

*Line 356. I think this should be 7% to 27%. If 27% have a LCB without a radar detection (at the location of cloud base) but 20% DO have a radar detection within 100 m, might it be the case that the radar is simply not able to see the bottom portion of the cloud and therefore only 7% of the time there is a "lidar only" detected cloud? (In my own work, I assume LCBs within 100m of radar detection are the same cloud and thus report a value closer to 10% for lidar-only detected clouds).

**Response:** This statement meant that 20% *of the* 27% of CEIL-only clouds did have identifiable reflectivity within 100 m above CEIL-identified CBH (therefore reducing CEIL-only cloud percentage from 27% to ~21% when considering these additional profiles). Figure R1 shows the fraction of CEIL-only clouds (which include only those with bases > 250 m) as a function of the distance to the nearest radar range gate with valid reflectivity ($Z_e$). The 27% of CEIL-only clouds decreases to 21% when accounting for any hydromteor-containing range gate within 100 m of the CEIL-identified CBH. While the CEIL-only fraction does converge to ~10%, this does not occur until the nearest reflectivity range gate is very far removed from the CEIL-identified CBH. The higher fraction of CEIL-only clouds in this dataset may be due to our retention of the BASTA time grid of 12 seconds, likely leaving more room for these occurrences to exist. It is, however, worth mentioning that the bottom portion of the cloud in these cases may just not be detectable

by the radar immediately at cloud base due to the small size of the droplets, for which we have also added discussion.

Clarification was added on lines 379-381 of the revised manuscript: "Here, the CEIL-only percentage reduces to ~ 20 % when also considering profiles where radar reflectivities exceed the noise floor within 100 m above LCB (not shown), which is evident of cloud layers where droplets are too small to be recognizable by BASTA at cloud base but become detectable as they grow above cloud base."

[Figure]

**Figure R1.** Fraction of total clouds with bases > 250 m AGL that do not have coincident reflectivity at ceilometer-identified CBH (i.e., CEIL-only clouds) as a function of the distance to the nearest radar range gate with valid reflectivity.

Line 394. The Comstock relationship was not developed using fog, which I believe is known to be composed of droplets that are smaller on average than that found for most clouds. I am not sure this is well justified.

**Response:** This is a fair point, and we believe our intention for saying that a large fraction of these near-surface clouds are experiencing precipitation from above can be inferred from the near-surface reflectivity averages anyway. We have therefore modified Fig. 8 to remove surface rain rate retrievals and modified the text on lines 434-437 of the revised manuscript stating the following: "Note that ~ 60 % of the distributions have $\overline{Z_{e,150-250m}}$ > -15 dBZ, suggesting a non-negligible portion of these near-surface clouds experience precipitation from above, for example as demonstrated in Fig. F1."

Line 503. Too much stratocumulus? I would say this is debatable, since the study in question is based on ISCCP obs and ISCCP-clustering. (1) While there is too much of the "stratocumulus" cluster, I am not sure there is too much actual stratocumulus cloud (in total) and (2) one needs to be careful with ISCCP measures for low cloud occurrence

(such is lower in ISCCP as compared with say MISR or CALIPSO/CloudSat). Perhaps change to "… and MAY now simulate …".

**Response:** This is a good point. We have modified the text to state the following on lines 544-545 of the revised manuscript: "Conversely, some CMIP6 models improved this bias and based on a classification of ISCCP data now may simulate too much stratocumulus that are not reflective enough (e.g., Schuddeboom et al., 2021)."

Line 505. "… 57% in GISS-ModelE3 compared to 76% in MICRE". So is it fair to conclude that cloud occurrence (and presumably stratocumulus) remains a bit low in GISS-ModelE3 (in spite of the previous statement)? Perhaps add a direct comment rather than leave such for the reader to piece together.

**Response:** This is indeed something we meant to highlight in the original draft. We have revised the sentence spanning lines 545-547 of the revised manuscript as follows: "In the current study, the occurrence frequency of liquid-based clouds is 57% in GISS-ModelE3 compared to 76% in MICRE (with month-to-month variability of ~ 10 percentage points), implying that GISS-ModelE3 cloud occurrence frequency is lower than observed."

Line 531. "if satellite observations underestimate precipitation occurrence frequency relative to collocated ground-based observations … " Don't the next two sentences demonstrate that this "middle point" IS true. Perhaps you should break this sentence apart and comment on the degree to which each of these might be true? As-is, you give three reasons, and say nothing about two of them, leaving the reader to infer what they will.

**Response:** We have restructured this paragraph (spanning lines 571-594 of the revised manuscript) to better address each of these points as follows:

"Here, we find that warm clouds precipitate *less* frequently in GISS-ModelE3 relative to ground-based observations, which is inconsistent with literature consensus based on satellite observations. Such differing conclusions could arise for several reasons. First, we demonstrated the likelihood that satellite observations underestimate precipitation occurrence frequency relative to colocated ground-based observations. Fig. 2h showed $P_{cb}$ for all liquid-based clouds using the sensitivity and vertical resolution of BASTA and for CloudSat 2C-PC "certain" and "possible" products, where $P_{cb}$ decreased from ~70 % for BASTA to ~35 % ("possible") and 20 % ("certain") for 2C-PC. Although the sensitivity and vertical resolution of CloudSat suggested by Fig. 2h does not account for CloudSat's data characteristics below 750 m AGL, this is roughly consistent with Tansey et al. (2022, see their Fig. 10), who showed that liquid-phase surface precipitation frequency decreased by ~30 % in their ground-based dataset compared to CloudSat. This comparison also implies that the GISS-ModelE3 $P_{cb}$ of 55 % could be larger than CloudSat suggests, but confirming that would require applying EMC[2] to GISS-ModelE3 outputs with CloudSat rather than BASTA radar characteristics. Related to this point, established model-observation comparisons may consider substantially different conditions owing to

sampling or methodology in general. For example, true cloud base is very difficult to observe from space-borne instrumentation, making cloud and precipitation somewhat ambiguous. Moreover, satellite studies have often focused on warm rain processes (Suzuki et al., 2015; Jing et al., 2017; Mülmenstädt et al., 2021). Clouds with CTTs > 0 ˚C during MICRE accounted for a smaller fraction of the cloud population than supercooled clouds, and most often warm cloud bases were below CloudSat's 750 m AGL threshold. Despite this, Kay et al. (2018) found that Southern Ocean supercooled cloud layers also produced snow too often in CESM1 relative to satellite observations, in contrast to our results. This leaves open the possibility that GISS-ModelE3 behaves differently from other ESMs, which could be verified by evaluating supercooled Southern Ocean clouds across multiple models to determine the prevalence of this reasoning. Reconciling these differing conclusions regarding ESM precipitation occurrence to which model results are sensitive (Mülmenstädt et al., 2021) will motivate further work to robustly evaluate models simultaneously against both ground-based observations and satellite observations, while directly comparing ground-based and space-based observations as demonstrated by Tansey et al. (2022). Additionally, ESM evaluation methodology using ground-based versus space-based simulators is worthy of further investigation since results and conclusions drawn can be sensitive to the representation of model physics (e.g., Cesana et al., 2021).”

Line 671. "...and 1 km"? I'm confused by this comment. Only near h_min is the sensitivity this good. Near 1 km you are near -36 dBZ, so a radar volume with -45 dBZ worth of precipitation at 1 km wont be detected.

**Response:** This statement was indeed misleading. The modifications made to address the “Line 219” comment above make this more clear.

Line 673. "underestimate", meaning the vendor LCB is too low in altitude (units are given in m not Pa)? I think you mean the CEIL LCB tends to be high !?! Perhaps rephrase to make clear.

**Response:** The original statement was incorrect. We have changed to “overestimate” on line 726 of the revised manuscript.

**Reviewer 2**

I commend the authors for producing a well-written manuscript. The analysis is solid, well-documented, and it characterizes cloud microphysical structure in a region of the planet, the Southern Ocean, where detailed observations are extremely rare. I have only a few minor comments and believe this manuscript is publishable in present form.

We thank Dr. Mark Miller for his review and suggestions to better explain/expand upon several physical understandings that were not previously considered. This has helped to extrapolate the techniques employed to further the understanding of the cloud types examined and physical processes operating within them.

Line 230: I was a bit curious about the justification for selecting the minimum Doppler velocity for Dmin rather than some measure of central tendency. Although you refer the reader to Silber (2021), a sentence outlining the reasoning behind this choice in the current manuscript would help the reader.

**Response:** This is done by considering that ice sublimates (or drifts from the radar's field of view) below cloud base more often than it grows below cloud base. Therefore, averaging the (reflectivity-weighted) Mean Doppler velocity across $D_{min}$ can provide a weak signal and very small precipitation rates. In this way, the minimum Mean Doppler velocity is viewed as a central upper limit to the precipitation rate. We have added the following language to clarify this on lines 247-249 of the revised manuscript: "The minimum mean (reflectivity-weighted) Doppler velocity is used as a central upper limit to the precipitation rate since preferential ice sublimation below LCB can significantly reduce precipitation rates when averaged across $D_{min}$."

Line 280: Is it possible that clouds that are part warm and part supercooled are frontal? My comment is more of a curiosity than substantive.

**Response:** Indeed, this does appear to often be the case. We didn't perform any formal cyclone/frontal analyses, but as shown in Fig. 3, partially supercooled clouds are thicker than purely warm or supercooled clouds. Time-height series of radar reflectivity for these profiles show structures consistent with warm and cold frontal passages.

Line 295: The maximum growth rate of ice is -14C, so I wasn't surprised that this difference exists. It may be worth mentioning this temperature for those who are not familiar with mixed phase clouds.

**Response:** We did mention that this temperature was the peak of vapor depositional growth, but have altered the following sentence slightly to more explicitly make this connection on lines 315-317 of the revised manuscript: "The precipitating fraction as a function CTT has a notable peak ~ -15 ˚C, which may be due to temperatures ~ -14 ˚C being the peak of vapor depositional

growth rates on ice (e.g., Fukuta and Takahashi, 1999; Wallace and Hobbs, 2006) increasing the likelihood of radar detectability, as also seen in Silber et al. (2021). "

Line 304: It would be interesting to characterize the vertical mixing state to determine the degree of decoupling (if there is any decoupling). There is a possibility that the cloud morphology you are describing is a hybrid: cumulus-coupled stratocumulus.

**Response:** This is a good point. We had already calculated the estimated inversion strength (EIS; Wood and Bretherton, 2006) using soundings. Figure R2 shows the average EIS as a function of (a) cloud thickness bins and (b) cloud top temperature (CTT) bins, as well as the precipitating fraction and median cloud-base precipitation rate ($R_{cb}$) as a function of EIS bins (c and d, respectively). Analysis here is limited to profiles within 3 hours of any given sounding. Figs. R1a and R1b show that EIS generally decreases with colder and thicker clouds (besides the thickest clouds > ~3-4 km, which are a small percentage of the population). In particular, partially supercooled clouds (orange lines) tend to produce the smallest EIS values as a function of CTT, while warm-only clouds produce the highest EIS values. This indicates that partially supercooled clouds are indeed more decoupled from the boundary layer than warm or even entirely supercooled clouds.

Lamer et al. (2020) evaluated the relationship between EIS and precipitation occurrence and intensity at the ARM Eastern North Atlantic (ENA) site, but only in subsidence regimes. They found that clouds more often precipitated with higher EIS with an upper EIS limit of 7-8 K. Figure R2c shows that for EIS values up to 7–8 K, precipitating fraction does indeed increase for purely warm and purely supercooled clouds, but precipitating fraction decreases with higher EIS. Lamer et al. (2020) also showed that cloud-base precipitation rates decrease slightly for higher EIS values, which is consistent with supercooled cloud bases here, but not for warm clouds (Fig. R2d).

We feel that a more formal analysis of boundary layer stability is beyond the scope of this study, but do think that a brief mention of these partially supercooled clouds tending to have lower EIS values is valuable and have added the following discussion to lines 326-328 in the revised manuscript: "Using soundings to calculate the estimated inversion strength (EIS; Wood and Bretherton, 2006), partially supercooled cloud layers were found to occur in environments associated with lower EIS values, indicating greater decoupling from the surface for this cloud type (not shown)."

We also refer to Truong et al. (2020) for a more thorough analysis of boundary layer climatology over the Southern Ocean using multiple field campaigns (including MICRE).

[Figure]

**Figure R2.** (a) Mean estimated inversion strength (EIS) as a function of cloud thickness (bin width = 200 m). (b) Mean EIS as a function of CTT (bin width = 2 °C). (c) Precipitating fraction as a function of EIS (bin width = 2 K). (d) Median cloud-base precipitation rate ($R_{cb}$) as a function of EIS (bin width = 2 K).

Line 515: Interesting finding. I wonder if precipitation occurence is really the best way to guage model performance. It is the vertical liquid/ice water flux that determines cloud water evolution. In other words, a cloud can precipitate almost continuously, but if the liquid/ice water flux is small, it may have minimal impact on the cloud life cycle. It's diffult to determine these fluxes from observations, as you note in the paper, but I wonder if there is really a difference in these fluxes between the cloud types despite the difference in precipitation occurence?

**Response:** We believe precipitation occurrence can indeed be a powerful constraint on model performance. Mülmenstädt et al. (2021) argue that, for warm clouds, identifying the presence of precipitation can be a proxy to a binary estimate of the autoconversion process, which is parameterized in models in a manner that produces a process rate. Nonetheless your argument stands, since the precipitation flux PDF can be a powerful constraint as well, as discussed in Silber et al. (2021). For supercooled clouds in particular, the sub-cloud supersaturation profile plays a significant role in determining the cloud-base precipitation rate PDF. By establishing these constraints observationally (accepting uncertainties) and pursuing process-oriented modeling studies, potential biases can be targeted. In an ESM framework, this provides the

ability to learn about processes impacting cloud-radiation feedbacks over the Southern Ocean that have a strong control on the equilibrium climate sensitivity (Zelinka et al., 2020). We have added the following statement on lines 708-710 of the revised manuscript to better emphasize this point: "Indeed, Mülmenstädt et al. (2021) argue that, for warm clouds, identifying the presence of precipitation can be a proxy to a simple binary estimate of the autoconversion process, which is parameterized in models in a manner that produces a process rate."

**References**

Hopkin, E., Illingworth, A. J., Charlton-Perez, C., Westbrook, C. D., and Ballard, S.: A robust automated technique for operational calibration of ceilometers using the integrated backscatter from totally attenuating liquid clouds, Atmos. Meas. Tech., 12, 4131–4147, https://doi.org/10.5194/amt-12-4131-2019, 2019.

Lamer, K., Naud, C. M., & Booth, J. F. (2020). Relationships between precipitation properties and large-scale conditions during subsidence at the Eastern North Atlantic observatory. Journal of Geophysical Research: Atmospheres, 125. https://doi.org/10.1029/2019JD031848

Mülmenstädt, J., Salzmann, M., Kay, J.E. et al. An underestimated negative cloud feedback from cloud lifetime changes. Nat. Clim. Chang. 11, 508–513 (2021). https://doi.org/10.1038/s41558-021-01038-1

O'Connor, E. J., A. J. Illingworth, and R. J. Hogan, 2004: A Technique for Autocalibration of Cloud Lidar. J. Atmos. Oceanic Technol., 21, 777–786, https://doi.org/10.1175/1520-0426(2004)021<0777:ATFAOC>2.0.CO;2.

Silber, I., Fridlind, A. M., Verlinde, J., Ackerman, A. S., Cesana, G. V., and Knopf, D. A.: The prevalence of precipitation from polar supercooled clouds, Atmos. Chem. Phys., 21, 3949–3971, https://doi.org/10.5194/acp-21-3949-2021, 2021.

Truong, S.C.H., Huang, Y., Lang, F., Messmer, M., Simmonds, I., Siems, S.T. and Manton, M.J. (2020), A Climatology of the Marine Atmospheric Boundary Layer Over the Southern Ocean From Four Field Campaigns During 2016–2018. J. Geophys. Res. Atmos., 125: e2020JD033214. https://doi.org/10.1029/2020JD033214

Wood, R., and C. S. Bretherton, 2006: On the Relationship between Stratiform Low Cloud Cover and Lower-Tropospheric Stability. J. Climate, 19, 6425–6432, https://doi.org/10.1175/JCLI3988.1.

Zelinka, M. D., Myers, T. A., McCoy, D. T., Po-Chedley, S., Caldwell, P. M., Ceppi, P., et al. (2020). Causes of higher climate sensitivity in CMIP6 models. Geophysical Research Letters, 47, e2019GL085782. https://doi.org/10.1029/2019GL085782